# *Lacticaseibacillus rhamnosus* P118 enhances host tolerance to *Salmonella* infection by promoting microbe-derived indole metabolites

**Baikui Wang[1†], Xianqi Peng[2,3†], Xiao Zhou[4], Xiuyan Jin[3], Abubakar Siddique[3], Jiayun Yao[2], Haiqi Zhang[2], Weifen Li[5], Yan Li[3,6], Min Yue[1]\***

[1]Key Laboratory of Systems Health Science of Zhejiang Province, School of Life Science, Hangzhou Institute for Advanced Study, University of Chinese Academy of Sciences, Hangzhou, China; [2]Zhejiang Institute of Freshwater Fisheries, Ministry of Agriculture and Rural Affairs Key Laboratory of Healthy Freshwater Aquaculture, Key Laboratory of Fish Health and Nutrition of Zhejiang Province, Key Laboratory of Fishery Environment and Aquatic Product Quality and Safety of Huzhou City, Huzhou, China; [3]Department of Veterinary Medicine, Zhejiang University College of Animal Sciences, Hangzhou, China; [4]Ningbo Academy of Agricultural Sciences, Ningbo, China; [5]Institute of Animal Nutrition and Feed Sciences, Zhejiang University College of Animal Sciences, Hangzhou, China; [6]Hainan Institute of Zhejiang University, Sanya, China

**\*For correspondence:**
myue@ucas.ac.cn

[†]These authors contributed equally to this work

**Competing interest:** The authors declare that no competing interests exist.

## eLife Assessment

The microbiome field is constantly providing insight on various health-related properties elicited by the commensals that inhabit their mammalian hosts. Harnessing the potential of these commensals for knowledge about host–microbe interactions, as well as properties with therapeutic implications, will likely remain a fruitful field for decades to come. In this **valuable** study, Wang et al. use various methods, encompassing classic microbiology, genomics, chemical biology, and immunology, to identify a potent probiotic strain that protects nematode and murine hosts from *Salmonella enterica* infection. The authors provide **compelling** evidence identifying gut metabolites that are correlated with protection and show that a single metabolite can recapitulate the effects of probiotic administration.

**Abstract** *Salmonella* is one of the most common foodborne pathogens, resulting in inflammatory gastroenteritis and frequently accompanied by dysbiosis. Gut commensals, such as *Lactobacillus* species, have been proven to exhibit broad antibacterial activities and protect hosts against pathogenic infections. Here, *Lacticaseibacillus rhamnosus* strain P118, screened from 290 isolates recovered from fermented yogurts and healthy piglet intestines using traditional and *Caenorhabditis elegans*-infection screening strategies, exerts great probiotic properties. Notably, P118 and its supernatant exhibited great antibacterial activities and attenuated *C. elegans* susceptibility to *Salmonella* infection. We found that P118 protected mice against *Salmonella* lethal infections by enhancing colonization resistance, reducing pathogen invasion, alleviating intestinal pro-inflammatory response, and improving microbial dysbiosis and fecal metabolite changes. Microbiota and fecal metabolome analyses suggested P118 administration significantly decreased the relative abundances of potentially harmful microbes (e.g., *Salmonella*, *Anaeroplasma*, *Klebsiella*)

and increased the fecal levels of tryptophan and its derivatives (indole, indole-3-acrylic acid, 5-hydroxytryptophan, 5-methoxyindoleacetate). Deterministic processes determined the gut microbial community assembly of P118-pretreated mice. Integrated omics further demonstrated that P118 probiotic activities in enhancing host tolerance to *Salmonella* infection were mediated by microbe-derived tryptophan/indole metabolites (e.g., indole-3-acrylic acid, indole, trypto-phan, 5-methoxyindoleacetic acid, and 5-hydroxytryptophan). Collective results demonstrate that *L. rhamnosus* P118 could enhance host tolerance to *Salmonella* infections via various pathways, including direct antibacterial actions, inhibiting *Salmonella* colonization and invasion, attenuating pro-inflammatory responses of intestinal macrophages, and modulating gut microbiota mediated by microbe-derived indole metabolites.

## Introduction

Diarrheal diseases caused by infectious agents (e.g., pathogens and parasites) remain a severe health burden worldwide. Approximately 1.7 billion childhood diarrhea cases are recorded annually, with diarrhea responsible for over 480,000 deaths for children aged <5 years and over 500,000 deaths for adults aged >70 years each year (*Li et al., 2022*; *Wang et al., 2024*; *Chen et al., 2024*). As one of the most common foodborne enteric pathogens, *Salmonella* infection leads to substantial gastro-enteritis incidents, imposing enormous economic burdens on global society and posing dispropor-tionate threats to animal and human health (*Stanaway et al., 2019*; *Wang et al., 2024*; *Zhou et al., 2025*). *Salmonella* has evolved strategies to subvert colonization resistance conferred by intestinal commensals and evade host immune defense responses, contributing to the increasing incidence of Salmonellosis (*Caballero-Flores et al., 2023*; *Feng et al., 2023*). *Salmonella*-contaminated food (meat, eggs, dairy) is considered the leading cause of Salmonellosis, estimated to cause a total global economic loss of over $3.5 billion annually in the US (*Stanaway et al., 2019*; *Jia et al., 2023*). Tradi-tionally, antibiotic treatment has been the primary strategy to control *Salmonella* infectious diseases (*Zhou et al., 2025*; *Jia et al., 2023*). However, the global abuse of antibiotics has raised significant concerns about antimicrobial resistance (AMR), foodborne antibiotic residues, and compromises in treating antimicrobial-resistant bacterial infections (*Wang et al., 2025*; *Jia et al., 2025*). These issues have inspired interest in seeking alternative strategies, such as dietary interventions (e.g., probiotics, prebiotics, natural products), to prevent *Salmonella* infectious disease outbreaks.

Microbial dysbiosis induced by antimicrobials, enteric infections, and irregular dietary habits increases the susceptibility to pathogen infection and other diseases, such as inflammatory bowel disease (*Gillis et al., 2018*; *Levy et al., 2017*). Therefore, a balanced intestinal microbiota and meta-bolic functions are crucial for maintaining gastrointestinal homeostasis. Gut commensal microbes and their derivatives have been proven to exhibit broad antimicrobial activities and protect the host against pathogen infections by providing colonization resistance against enteric pathogens, limiting pathogen colonization, invasion, and transmission, and maintaining intestinal barrier function (*Caballero-Flores et al., 2023*; *Jacobson et al., 2018*; *Yaqoob et al., 2024*). As one of the intestinal commensal microbes, probiotic species (without residues in raw food products), such as *Lactobacillus*, *Bifidobac-terium*, *Bacillus,* and yeast (e.g., *Saccharomyces boulardii*, *S. cerevisiae*), exert beneficial effects on the host by enhancing colonization resistance and immune defense, inhibiting colonization and invasion against pathogens, and maintaining gastrointestinal homeostasis (*Han et al., 2024*; *Zhang et al., 2024*). Although many candidate probiotic isolates have been uncovered from fermented foods and mammalian intestines (*Sakandar and Zhang, 2021*; *Wang et al., 2022*), the traditional strategies for screening these candidates are both time-consuming and labor-intensive, involving bacterial isolation, culturing, phenotypic characterization, randomized controlled trials, and various in vitro and in vivo tests to assess probiotic properties (*Sun et al., 2022*). While culture-dependent methods are classic strategies, emerging evaluation strategies for candidates, such as whole-genome sequencing-based approaches and small invertebrate models (e.g., *Caenorhabditis elegans*, *Drosophila*), have gained significant attention due to their high-throughput, replicable, and standardized properties (*Sun et al., 2022*; *Wang et al., 2022*).

Intestinal macrophages play vital roles in maintaining gut homeostasis, regulating inflammation, and particularly in promoting the resolution of inflammation (*Lavelle and Sokol, 2020*; *Na et al., 2019*), which has been considered a novel potential target for controlling intestinal pro-inflammatory

disorders. It has been reported that probiotics-derived metabolites, such as bacteriocins, antimicrobial compounds, short-chain fatty acids, and tryptophan/indole derivatives, can mediate the beneficial effects of probiotics on the host health by interacting with the host gastrointestinal immune cells and gut residents (*Britton and Faith, 2021*; *Sanders et al., 2019*). Bacterial tryptophan metabolism produces indole derivatives (e.g., indole-3-acetic acid, 3-indolepropionic acid, 3-indoleacrylic acid, indole-3-lactic acid, indole-3-aldehyde, indole-3-acetaldehyde), which are potent bioactive ligands for the aryl hydrocarbon receptor (AHR) (*Agus et al., 2018*; *Koh and Bäckhed, 2020*). These indole derivatives play crucial roles in maintaining gut barrier integrity and homeostasis by activating AHR signaling pathway (*Agus et al., 2018*). *Limosilactobacillus reuteri*-produced indole derivatives have been reported to exert anti-inflammatory effects by activating AHR signaling pathway (*Cervantes-Barragan et al., 2017*). Although the beneficial effects of probiotics in protecting the host against diverse enteric infections have been extensively examined, the interactive roles between pro-inflammatory macrophages and microbe-tryptophan/indole metabolites in combating enteric infections have not been fully studied yet. Here, we found that *Lacticaseibacillus rhamnosus* P118 exerts great probiotic properties after assessing by traditional and *C. elegans*-infection screening strategies. P118 exhibited broad antibacterial activities and reduced host susceptibility to enteric *Salmonella* infection by improving intestinal dysbiosis and fecal metabolite changes, and inhibiting intestinal pro-inflammatory responses. Integrated omics further demonstrated that P118 probiotic activities in enhancing host tolerance to *Salmonella* infection were mediated by microbe-derived tryptophan/indole metabolites.

## Results

### Two screening approaches to convergent probiotic candidates

After being identified by Matrix-Assisted Laser Desorption Ionization-Time of Flight Mass Spectrometry (MALDI-TOF MS), a total of 290 bacterial isolates were isolated and identified from 33 fermented yogurt samples and 6 healthy piglet rectal content samples. Those isolates consist of 63 *Streptococcus* isolates, 158 *Lactobacillus/Lacticaseibacillus/Limosilactobacillus* isolates, and 69 *Enterococcus* isolates (*Figure 1A*, *Table 1*). Two screening strategies were employed in the present study to further investigate the potential probiotic properties of the isolates: the traditional/classic approach and the *C. elegans* infection approach. In the traditional/classic approach, 27 isolates were screened out by milk-clotting activity assay (*Figure 1A*, *Supplementary file 1*), among which two isolates (P118 and P199) exhibited the highest tolerance to bile salt (0.3–2.0%) (*Figure 1B*) and biofilm formation capabilities (*Figure 1C*). Compared with the P199 strain, P118 strain exhibited the highest susceptibility to multiple antimicrobials (*Figure 1—figure supplement 1*). *C. elegans* has been widely used as an invaluable model for understanding the conserved mechanisms of host–microbe interactions due to the similarities of gut morphology and physiological function with human and animal (*Kumar et al., 2020*). In the *C. elegans* infection approach, 8 out of 290 isolates significantly increased the survivals and prolonged the life span of *S.* Typhimurium-infected *C. elegans* (p<0.05), and three isolates (P118, P119, P120)-treated worms were more resistant to *S.* Typhimurium infection than the other isolates (p<0.05) (*Figure 1D*, *Supplementary file 2*). Integrating the results of both screening approaches, *L. rhamnosus* P118 strain exhibits versatile capabilities as a probiotic candidate (*Figure 1E*).

To illustrate the probiotic properties of *L. rhamnosus* P118, antibacterial activity evaluation in vitro was further conducted, and the results showed that the fermented supernatants of P118 significantly inhibited the growth of multiple pathogens (e.g., *Salmonella*, *Yersinia*, *Staphylococcus aureus*, *Escherichia coli*, *Citrobacter rodentium*, *Pseudomonas aeruginosa*, *Riemerella anatipestifer*) (*Figure 1—figure supplement 1*). Interestingly, the inhibitory effect of P118 against *S.* Typhimurium was in dose- and oxygen-dependent manners, and results showed that high-dose and anaerobic cultures exhibited more antibacterial effects than low-dose and aerobic cultures, respectively (p<0.05) (*Figure 1F*). Active factors derived from P118 to exert antibacterial effects were explored. The results showed that the antibacterial activities of the fermented supernatant (pH=3.4) were sensitive to alkalinity (pH>5.0), trypsin, proteinase-K, pepsin, and high temperature (>70°C), but were resistant to catalase treatment (*Figure 1G and H*). The fermented cultures and supernatants of P118 not only significantly inhibited the growth of *S.* Typhimurium in vitro (*Figure 1I*), but also protected worms against *S.* Typhimurium infection (*Figure 1G*, *Supplementary file 3*). Although they exhibited no

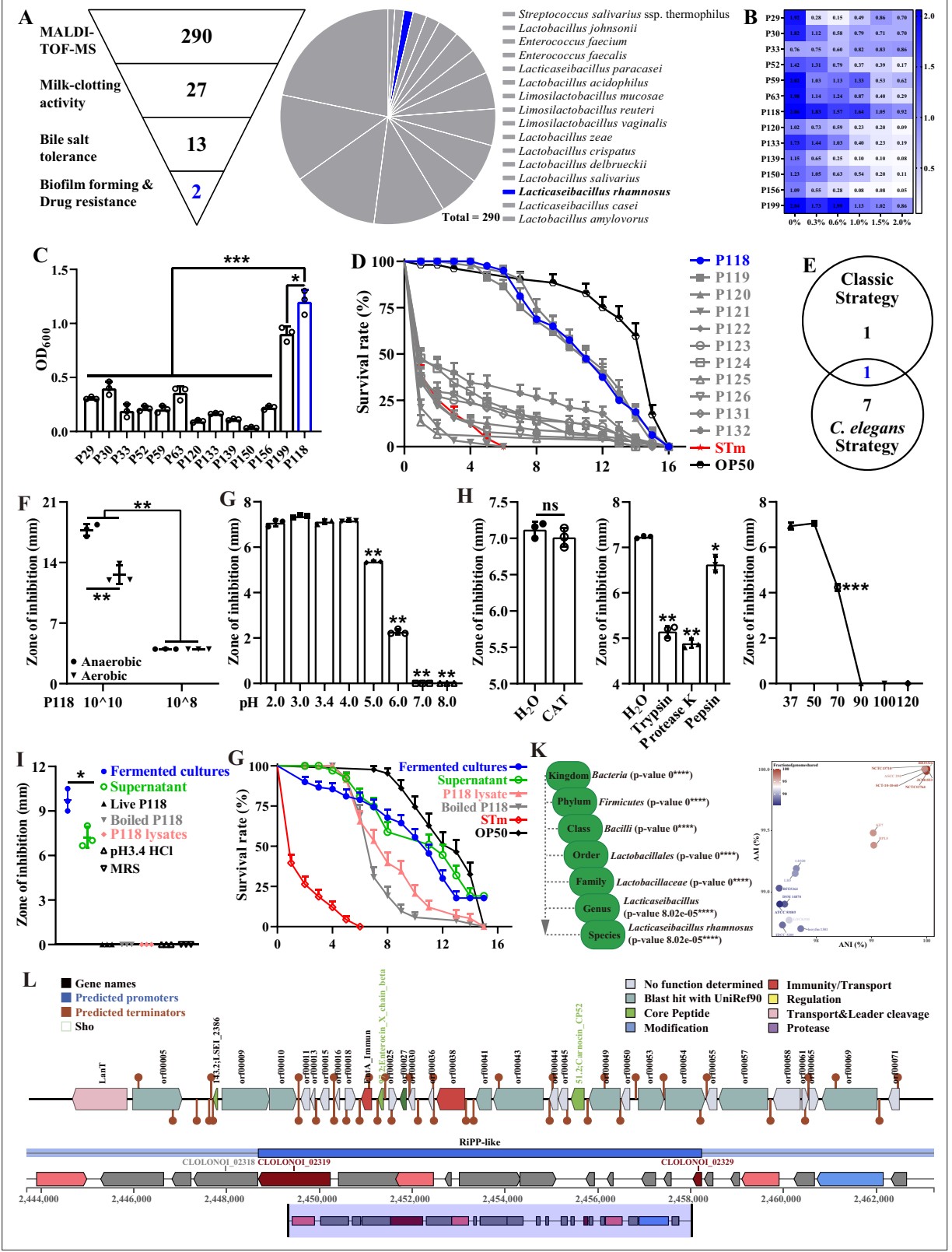

**Figure 1.** Isolation and antibacterial characterization of *L. rhamnosus* P118. (**A**) Screening flowchart of P118 in vitro and candidate probiotic isolates identified by MALDI-TOF MS. (**B**) Bile salt tolerance ability of the isolates. (**C**) Biofilm-forming ability of isolates. (**D**) Screening using *C. elegans* infection model. (**E**) Interacted screening strategy. (**F**) Antibacterial ability of P118 under aerobic or anaerobic culture conditions. (**G**) Broad-spectrum pH tolerance of P118 supernatants (pH = 3.4) that adjusted to pH <3.4 by 1 M HCl or pH >3.4 by 1 M NaOH. (**H**) Antibacterial effects of P118 supernatant

*Figure 1 continued on next page*

*Figure 1 continued*

under 20 mg/mL catalase (CAT), 100 µg/mL proteinases (trypsin, proteinase K, pepsin), and different temperatures (37, 50, 70, 90, 100, 120°C) boiled for 30 min treatments. (**I**) Antibacterial effects of components of P118 (boiled at 120°C for 30 min or was lysed by ultrasonication at 240 W for 2 h). (**J**) Active ingredients of P118 protect *C. elegans* against *S.* Typhimurium infection. (**K**) Taxonomic classification of P118 draft genome queried against the NCBI non-redundant prokaryotic genomes database with p-values representing confidence of phylogenetic assignment, and the nearest subspecies phylogenetic neighbor of P118 draft genome was determined by percentage shared genomic content graphed as ANI versus AAI. (**L**) Prediction of secondary metabolites and bacteriocin protein-encoding gene clusters of P118 using antiSMASH and BAGEL4 databases. (**F–J**) *S.* Typhimurium SL1344 was selected as an indicator pathogen. Significant differences *p<0.05, **p<0.01, ***p<0.001.

The online version of this article includes the following figure supplement(s) for figure 1:

**Figure supplement 1.** Drug resistance and antibacterial activities of *Lactobacillus* isolates.

**Figure supplement 2.** Genome map and taxonomic classification of *L. rhamnosus* P118.

antibacterial activities in vitro (***Figure 1I***), the protective activities of heat-treated P118 and P118 lysates against *S.* Typhimurium pathogenesis were observed in *C. elegans* but weaker than that of the fermented cultures and supernatants (***Figure 1G***).

## P118 genomics analysis suggests genetic determinants for antibacterial actions

Subsequently, whole-genome sequencing of P118 was performed to explore the potential antibacterial factors based on genomic evidence. The results showed that species-level identity for P118 isolate with 2.99 Mb genome size was confirmed at p<0.0001 (***Figure 1—figure supplement 2***, ***Figure 1K***, ***Supplementary file 4***), and P118 draft genome was most closely related to *L. rhamnosus* subspecies (NCTC13710 and BIO5326) (***Figure 1K***, ***Supplementary file 5***, ***Figure 1—figure supplement 2***). Genomic analysis using antiSMASH and BAGEL4 databases revealed the presence of putative bacteriocin synthesis genes in P118 isolate (***Figure 1L***). Putative bacteriocins predicated in P118 isolate showed high homology with known class II bacteriocins (IIa carnobacteriocin B2 produced by *C. maltaromaticum*, IIb enterocin X produced by *Enterococcus faecium* KU-B5, IId/b lactococcin A/G,

**Table 1.** A list of probiotic isolates recovered from examined samples.

| Source | Name | Number of isolates |
|---|---|---|
| | *Streptococcus salivarius* ssp. *thermophilus* | 63 |
| | *Lacticaseibacillus paracasei* | 18 |
| | *Lactobacillus acidophilus* | 17 |
| | *Lactobacillus zeae* | 10 |
| | *Lacticaseibacillus rhamnosus* | 4 |
| | *Lacticaseibacillus casei* | 4 |
| Yogurt | *Lactobacillus delbrueckii* | 6 |
| | *Limosilactobacillus mucosae* | 16 |
| | *Lactobacillus johnsonii* | 38 |
| | *Limosilactobacillus reuteri* | 16 |
| | *Limosilactobacillus vaginalis* | 11 |
| | *Lactobacillus crispatus* | 9 |
| | *Lactobacillus salivarius* | 6 |
| | *Lactobacillus amylovorus* | 3 |
| | *Enterococcus faecium* | 38 |
| Piglet intestine | *Enterococcus faecalis* | 31 |
| Total | | 290 |

and some novel bacteriocin-encoded hypothetical proteins). Collective data demonstrates that *L. rhamnosus* P118 exhibits outstanding probiotic traits.

## P118 protects lethal *S.* Typhimurium infections in a murine model

The protective activity of *L. rhamnosus* P118 against enteric pathogens was further examined in the *Salmonella*-infected murine model (*Figure 2A*). Administration of P118 significantly protected mice against *Salmonella*-induced deaths and body weight losses (*Figure 2B and C*), accompanied by the alleviated splenomegaly, hepatomegaly, shortened gastrointestinal tract, and clinical and pathological score (*Figure 2D–F*, *Figure 2—figure supplement 1*). Meanwhile, P118 significantly limited *Salmonella* colonization in the intestinal tissues (duodenum and colon) and invasion into the peripheral organs (liver and spleen) of mice, thereby markedly reducing *Salmonella* fecal shedding (*Figure 2G*). Additionally, *Salmonella* infection-induced ileal mucosal damage was alleviated by P118 administration, as illustrated by the increased ileal villus height (*Figure 2H*, *Figure 2—figure supplement 1*), ratio of villus height/crypt depth (*Figure 2H*, *Figure 2—figure supplement 1*), ileal microvilli height and density (*Figure 2H*, *Figure 2—figure supplement 1*), the upregulated *Tjp1* mRNA expression (*Figure 2—figure supplement 2*), and the downregulated *Il1b* mRNA expression (*Figure 2—figure supplement 2*), accompanied by the reduced Il1b level in serum (*Figure 2—figure supplement 2*). Intervention with P118 significantly increased the numbers of Ki67-positive cells (proliferating cells), Muc2-positive cells (goblet cells), and decreased the numbers of F4/80-positive cells (macrophages) and F4/80$^+$ Nos2$^+$ cells (pro-inflammatory macrophages) in the ileum (*Figure 2H*) and also attenuated the pathological damages of peripheral organs (liver and spleen) caused by *Salmonella* infection (*Figure 2—figure supplement 1A and B*).

## P118 improves *Salmonella* infection by modulating gut microbiota

Gut microbes play crucial roles in host physiological activities and colonization resistance against enteric pathogens (*Jacobson et al., 2018*), and dysbiosis is supposedly associated with *Salmonella* infection. Significant bacterial community structures among four groups were observed (*Figure 3A*, *Supplementary file 8*), and most of the bacterial operational taxonomic units (OTUs) were classified as group-specific OTUs (*Figure 3B*). Compared to the uninfected group, *Salmonella* infection significantly depleted 167 OTUs and enriched 60 OTUs (*Figure 3C*), whereas compared to the *Salmonella*-infected group, pretreatment with P118 significantly depleted 16 OTUs and enriched 52 OTUs (*Figure 3D*). Specifically, *Salmonella* infection significantly increased the relative abundances of potentially harmful microbes (e.g., *Salmonella*, *Escherichia-Shigella*, *Klebsiella*, *Morganella*, *Akkermansia*) and significantly reduced the relative abundances of SCFA-producing and potentially beneficial microbes (e.g., *Alloprevotella*, *Lactococcus*, *Faecalibacterium*, *Limosilactobacillus*, *Parasutterella*, *Odoribacter*, *Dubosiella*, *Candidatus arthromitus*) (*Figure 3E*, *Figure 3—figure supplement 1B*). Pretreatment with P118 significantly reduced the relative abundances of pathogenic microbes (e.g., *Salmonella*, *Anaeroplasma*, *Klebsiella*, *Morganella*) and significantly increased the relative abundances of potentially beneficial or commensal microbes (e.g., *Candidatus_Saccharimonas*, *Turicibacter*, *Enterorhabdus*) (*Figure 3F*, *Figure 3—figure supplement 1C*).

Based on the changes in bacterial community structures, the internal driving forces of gut microbial communities were investigated using ecological models. Both null model and neutral community model analyses revealed that the stochastic processes belonging to homogenizing dispersal exerted important roles in *Salmonella*-infected mice gut microbiome assembly, whereas the deterministic processes (homogeneous selection and variable selection) exerted more influences on P118-pretreated mice gut microbiome assembly than *Salmonella*-infected mice (*Figure 3—figure supplement 2A and B*).

## Microbe-derived tryptophan metabolites are associated with the protection

Significant differences in fecal metabolite structures among the three groups separated into distinct clusters were also observed (*Figure 4A*, *Supplementary file 9*). Compared with the uninfected group, *Salmonella* infection significantly upregulated 256 fecal metabolites (log$_2$(fold change)>1, p<0.05) and downregulated 402 fecal metabolites (log$_2$(fold change)<1, p<0.05) (*Figure 4B*), and pretreatment with P118 significantly enriched 100 metabolites (log$_2$(fold change)>1, p<0.05) and depleted

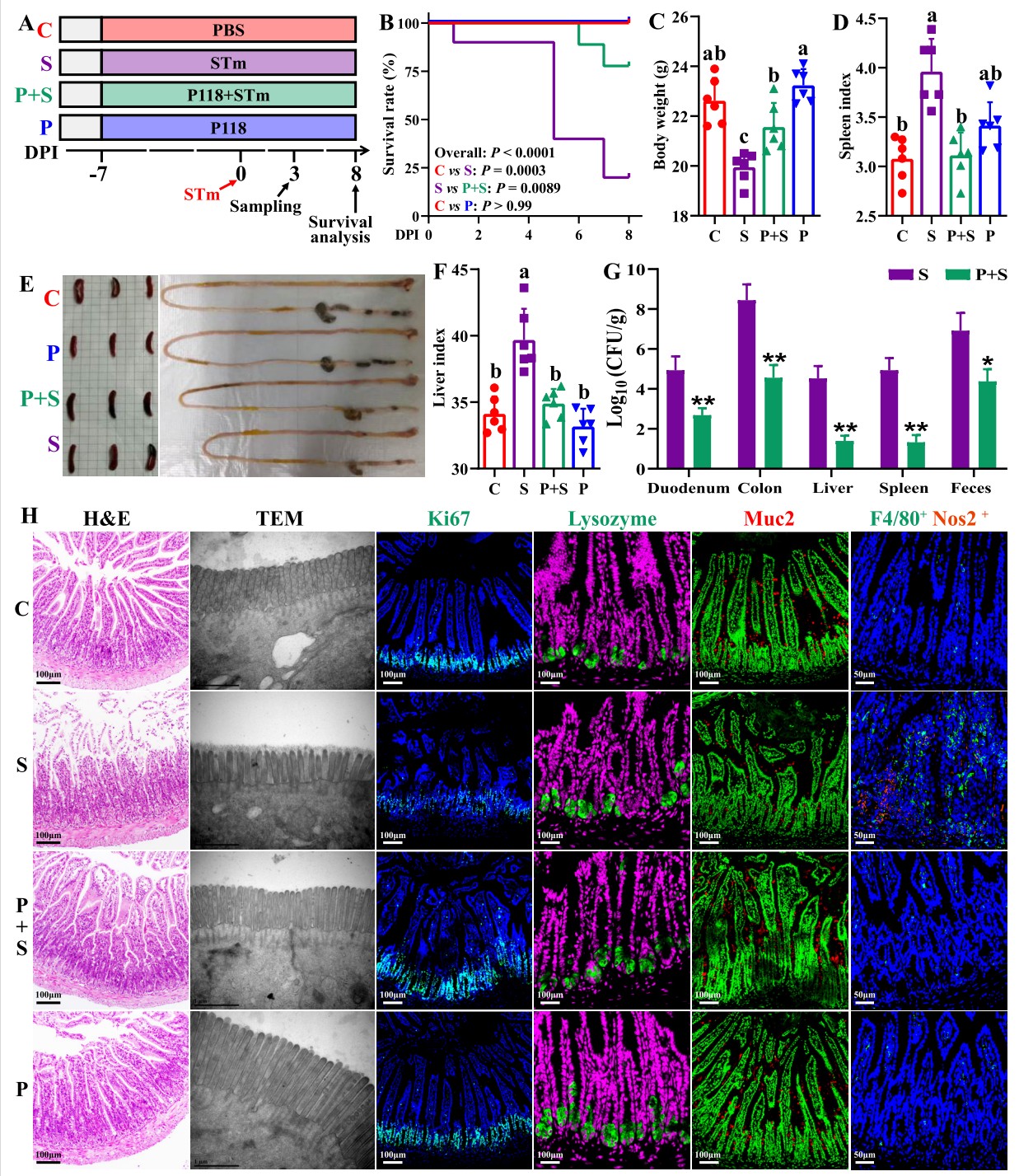

**Figure 2.** *L. rhamnosus* P118 enhances tolerance to *S.* Typhimurium infection in mice. (**A**) Experimental design. (**B**) Survival curve of mice infected with *S.* Typhimurium. (**C**) Body weight. (**D**) Spleen index. (**E**) Representative images of spleen and intestine. (**F**) Liver index. (**G**) *S.* Typhimurium burden in tissues and shedding in feces. (**H**) Representative images of H&E staining, TEM, and immunostaining (DAPI, Ki67, lysozyme, Muc2, F4/80, Nos2) in the ileum. Different lowercase letters indicate a significant difference (p<0.05). Significant differences *p<0.05, **p<0.01. C: PBS group; P: P118 administered group; S: *S.* Typhimurium-infected group; P+S: P118 protective group.

The online version of this article includes the following figure supplement(s) for figure 2:

**Figure supplement 1.** Clinical and histopathological observation.

**Figure supplement 2.** Gene (**A, B, E, F**) and protein (**C, D**) expression level of tight junction protein (*Tjp1*, *Claudin-1*) and inflammatory cytokines (Il1b, Il18) in ileum and serum of mice.

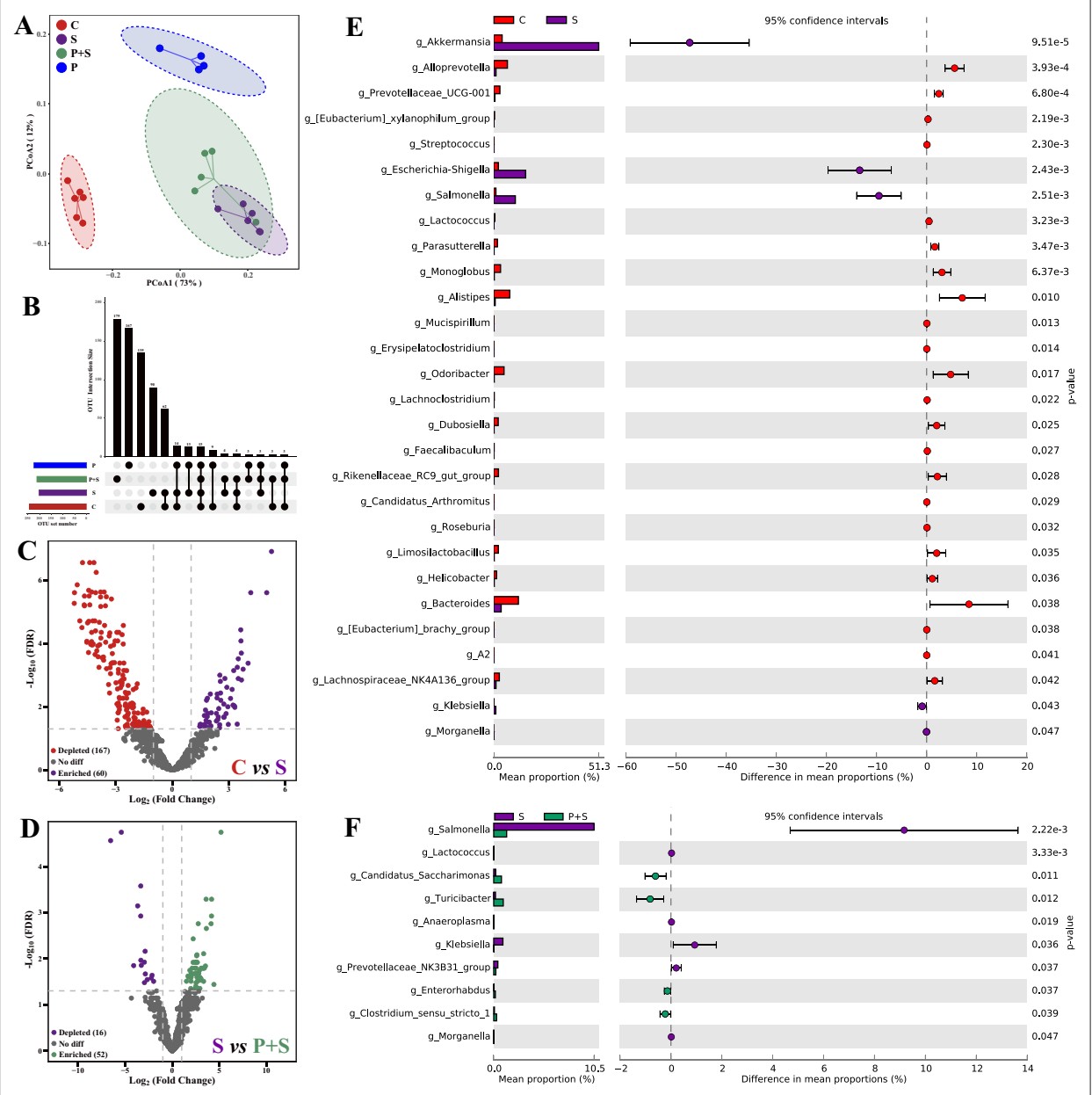

**Figure 3.** *L. rhamnosus* P118 improves *S.* Typhimurium infection-induced dysbacteriosis. (**A**) Principal coordinates analysis (PCoA) based on Bray-Curtis distance. (**B**) UpSetR plot based on bacterial absolute operational taxonomic unit (OTU) abundances. (**C, D**) The fold changes of bacterial absolute OTU abundances between two groups. (**E, F**) Comparison of intestinal microbes by STAMP. The prefix 'g_' represents the annotated level of the genus. C: PBS group; P: P118 administered group; S: *S.* Typhimurium-infected group; P+S: P118 protective group.

The online version of this article includes the following figure supplement(s) for figure 3:

**Figure supplement 1.** Comparison of intestinal microbes by the analysis of statistical analysis of taxonomic and functional profiles (STAMP) software with a two-sided Welch's *t*-test.

**Figure supplement 2.** *L. rhamnosus* P118 alters gut microbial community assembly.

81 metabolites (log$_2$(fold change)<1, p<0.05) (***Figure 4C***). Among the significant differential metabolites, 91 metabolites (VIP>1) downregulated in *Salmonella*-infected mice were upregulated explicitly in P118-pretreated mice (***Figure 4—figure supplement 1A and C***, ***Supplementary file 10***), and 43 metabolites (VIP>1) upregulated in *Salmonella*-infected mice were downregulated explicitly in P118-pretreated mice (***Figure 4—figure supplement 1B and C***, ***Supplementary file 10***). KEGG pathway analysis revealed that fecal metabolites were mainly enriched in tryptophan metabolism,

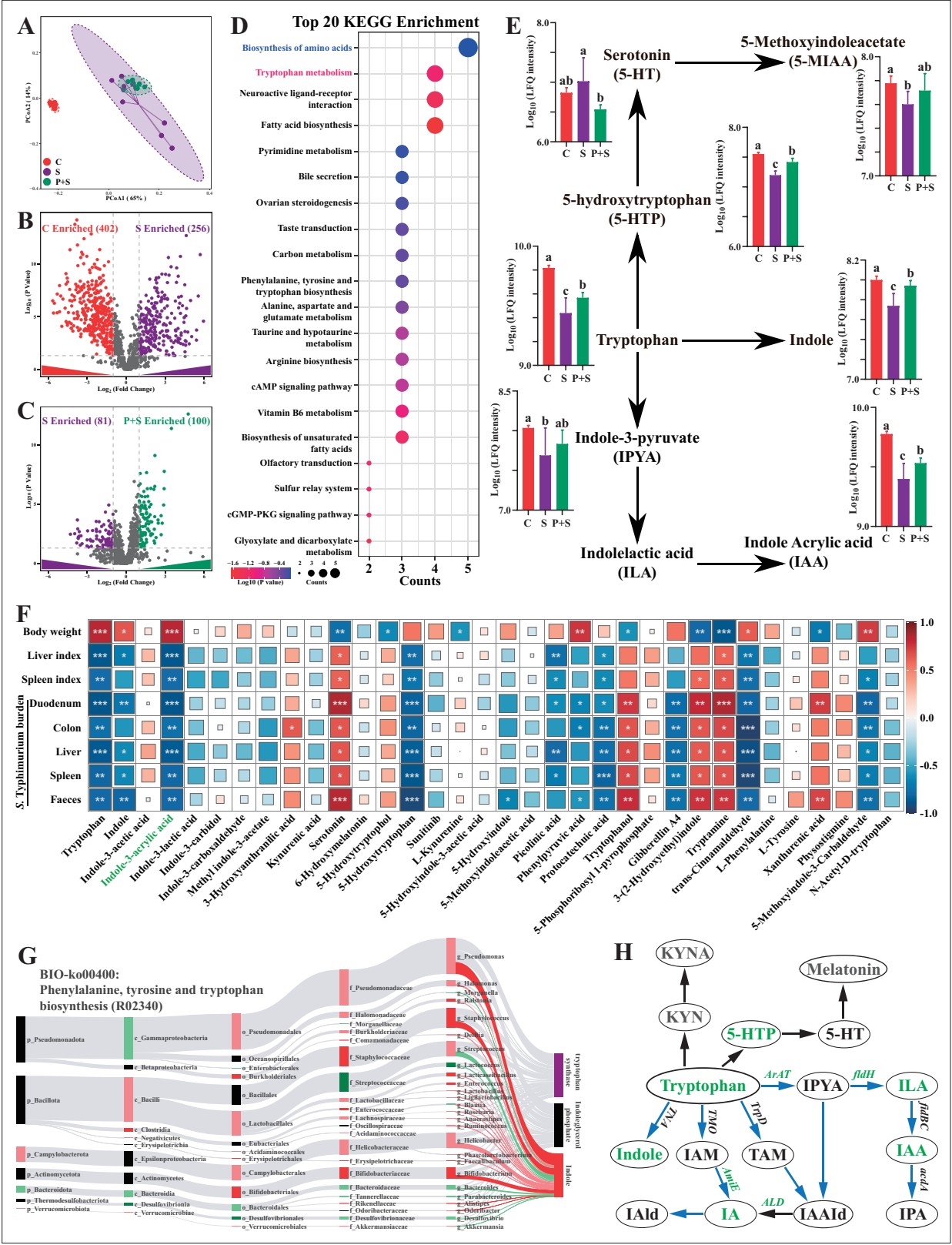

**Figure 4.** Microbe-derived tryptophan metabolites are involved in mice tolerance to *S.* Typhimurium. (**A**) Principal coordinates analysis (PCoA) based on Bray–Curtis distance of fecal metabolites. (**B, C**) UpSetR plot based on fecal metabolites. (**D**) Metabolomics pathway enrichment in 'P+S vs. S'. (**E**) Comparison of fecal microbial tryptophan metabolism-enriched pathway among groups. (**F**) Pearson correlation analysis among *S.* Typhimurium burden, organ indices, body weight, and fecal metabolites in mice. (**G**) Bio-Sankey network analysis between intestinal microbes and fecal metabolites.

*Figure 4 continued on next page*

*Figure 4 continued*

(**H**) Pathway schematic of abbreviated mammalian and microbial tryptophan metabolism. Different lowercase letters indicate a significant difference (p<0.05). Significant differences *p<0.05; **p<0.01; ***p<0.001. C: PBS group; P: P118 administered group; S: *S.* Typhimurium-infected group; P+S: P118 protective group.

The online version of this article includes the following figure supplement(s) for figure 4:

**Figure supplement 1.** Analysis of differential fecal metabolites.

fatty acid biosynthesis, biosynthesis of amino acids, and neuroactive ligand–receptor interaction pathways in P118-pretreated mice (*Figure 4D*). Interestingly, pretreatment with P118 significantly increased the fecal levels of tryptophan and tryptophan derivatives (indole, indole-3-acrylic acid [IAA], 5-hydroxytryptophan, 5-methoxyindoleacetate) (*Figure 4E*). Pearson correlation analysis showed that *S.* Typhimurium burdens (in the duodenum, colon, liver, spleen, feces) and organ (spleen, liver) indices were negatively correlated with the levels of fecal tryptophan and tryptophan derivatives (indole, IAA, 5-hydroxytryptophan, protocatechuic acid, *trans*-cinnamaldehyde, 5-methoxyindole-3-carbaldehyde) and positively correlated with the levels of (serotonin, tryptophanol, 3-(2-hydroxyethyl) indole, tryptamine) (*Figure 4F*). Conversely, the body weight was positively correlated with the levels of fecal tryptophan and tryptophan derivatives (*Figure 4F*). Bio-Sankey network analysis further showed that 26 genera were identified as potential bacteria that might participate in the metabolic reaction R02340 (tryptophan synthase, indoleglycerol phosphate, and indole), of which five bacteria (*Bifidobacterium*, *Lacticaseibacillus*, *Enterococcus*, *Staphylococcus*, and *Pseudomonas*, dark red if p<0.05) were positively associated with indole in the metabolic reaction R02340 (*Figure 4G*). These results indicate that microbe-derived tryptophan metabolites are associated with P118-mediated protection against *Salmonella* infection.

Based on the findings above that *Lacticaseibacillus* species were involved in tryptophan biosynthesis and metabolism, *L. rhamnosus* P118 genome was reanalyzed to determine whether P118 encodes enzymes necessary to exert this function. The genomic data further showed that P118 encoded various genes (e.g., *fldH*, *AraT*, *AspB-4*, *AmiE*, *trpA*, *trpB*, *BCAT*, *IGPS*, *MT*, *ALD*) essential to biosynthesize and metabolize tryptophan into tryptophan derivatives (*Figure 4H*, *Supplementary file 11*). The metabolomic results of P118 cultures also validated the genomic data that P118 could secrete a diverse profile of tryptophan-derived metabolites (e.g., IAA, indole, indole-3-lactic acid, DL-tryptophan, kynurenine, N-acetyl-d-tryptophan, 5-methoxyindoleacetic acid, 5-hydroxytryptophan) (*Supplementary file 12*). Taken together, these data suggest that P118-derived tryptophan metabolites might contribute to the protection against *Salmonella* infection.

## Indole-3-acrylic acid protects against *Salmonella* infection by inhibiting macrophage pro-inflammatory responses

Based on the above results, exogenous IAA was employed to further explore its roles against *Salmonella* infection. The antibacterial assay results showed that IAA (>4.4 mM) significantly inhibited *Salmonella* growth (*Figure 5—figure supplement 1*). Exogenous IAA administration significantly reduced mice susceptibility to *Salmonella* infection, as evidenced by the increased body weight (*Figure 5B*), the alleviated splenomegaly and hepatomegaly (*Figure 5D and F*), and the increased colon length (*Figure 5C*). Meanwhile, IAA administration significantly inhibited *Salmonella* colonization in the intestinal tissues (cecum and colon) and invasion into the peripheral organs (liver and spleen) of mice, thereby markedly reducing *Salmonella* fecal shedding (*Figure 5E*). Additionally, *Salmonella* infection-induced ileal mucosa damage was alleviated by exogenous IAA, as illustrated by the increased ileal villus height (*Figure 5G and I*), and the increased numbers of Muc2-positive goblet cells in the ileum (*Figure 5H and I*). Exogenous IAA intervention significantly reduced *Salmonella* infection-induced intestinal pro-inflammatory responses, as evidenced by the decreased numbers of F4/80-positive macrophages (*Figure 5I*), the downregulated mRNA (*Il1b*, *Il6*, *Il18*, *Tnfa*, *Nos2*) expression (*Figure 5H*), and the reduced protein (Nos2, Il1b, Il6, Tnfa, markers of pro-inflammatory macrophages) expression (*Figure 5I*).

The in vitro experiments further showed that, although they failed to increase the uptake of *Salmonella* (*Figure 5—figure supplement 2A*, 0 h), IAA treatment significantly reduced intracellular *Salmonella* survival in RAW 264.7 macrophage cells (*Figure 5—figure supplement 2A*, 6 h 12 h), which

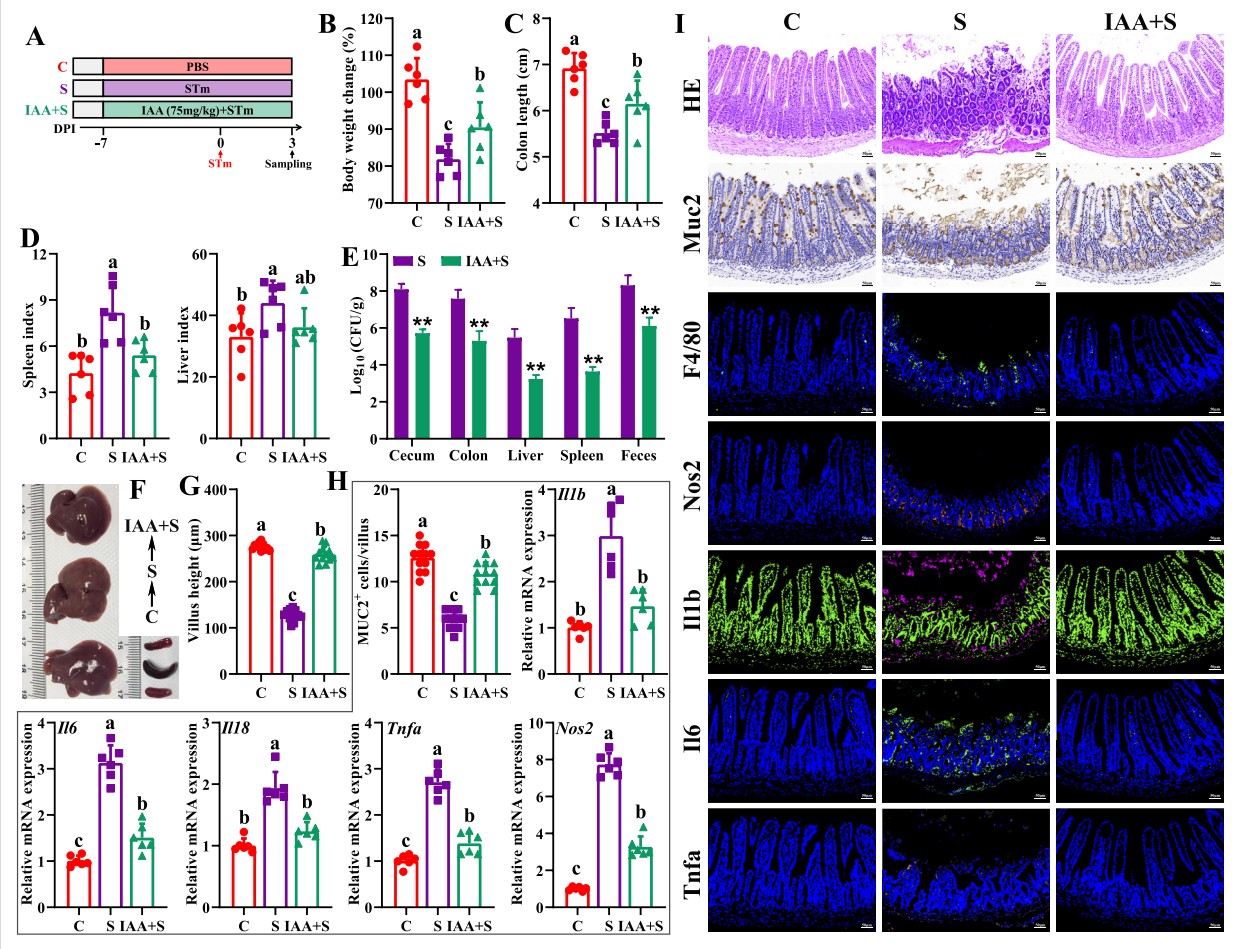

**Figure 5.** Indole-3-acrylic acid enhances mice tolerance to *S.* Typhimurium infection. (**A**) Experimental design. (**B**) Body weight change. (**C**) Colon length. (**D**) Spleen and liver indexes. (**E**) *S.* Typhimurium burden in tissues and shedding in feces. (**F**) Representative images of spleen and liver. (**G**) Villus height of ileum. (**H**) Muc2-positive cells and mRNA expression levels in ileum. (**I**) Representative images of H&E staining and immunostaining in the ileum. Different lowercase letters indicate a significant difference (p<0.05). C: PBS group; S: *S.* Typhimurium-infected group; IAA + S: indole-3-acrylic acid protective group.

The online version of this article includes the following figure supplement(s) for figure 5:

**Figure supplement 1.** Antibacterial activities of indole-3-acrylic acid.

**Figure supplement 2.** Indole-3-acrylic acid enhances bactericidal capacity and exerts anti-inflammatory activity in RAW 264.7 cells.

could be reversed by pretreatment with AHR inhibitor CH-223191 (*Figure 5—figure supplement 2*, 6 h, 12 h), indicating that IAA enhances the intracellular bactericidal capacity of RAW 264.7 cells. Furthermore, IAA treatment significantly inhibited *Salmonella* infection-induced pro-inflammatory responses, as evidenced by inhibiting mRNA (*Il1b*, *Il6*, *Il18*, *Tnfa*, *Nos2*) expression and nitric oxide secretion of *Salmonella*-infected RAW 264.7 macrophage cells, which was blocked by the presence of AHR inhibitor CH-223191 (*Figure 5—figure supplement 2B-G*).

Macrophages play important roles in initiating immune responses, maintaining gut homeostasis, and phagocytic clearance of pathogens (*Na et al., 2019*). To further investigate the involved roles of macrophages in P118/IAA-mediated protective effect, in vivo macrophage depletion using clodronate liposomes was conducted. As expected, macrophage depletion significantly blocked the protection of P118 and exogenous IAA against *Salmonella* infection (*Figure 6A–I*). P118 or exogenous IAA administration failed to increase the body weight (*Figure 6B*) and colon length (*Figure 6E*), alleviate the splenomegaly and hepatomegaly (*Figure 6C and D*), and inhibit the colonization, invasion, and shedding of *Salmonella* (*Figure 6F*) after the intestinal macrophage depletion. Additionally, after depleting the intestinal macrophage, P118 or exogenous IAA administration also failed to attenuate

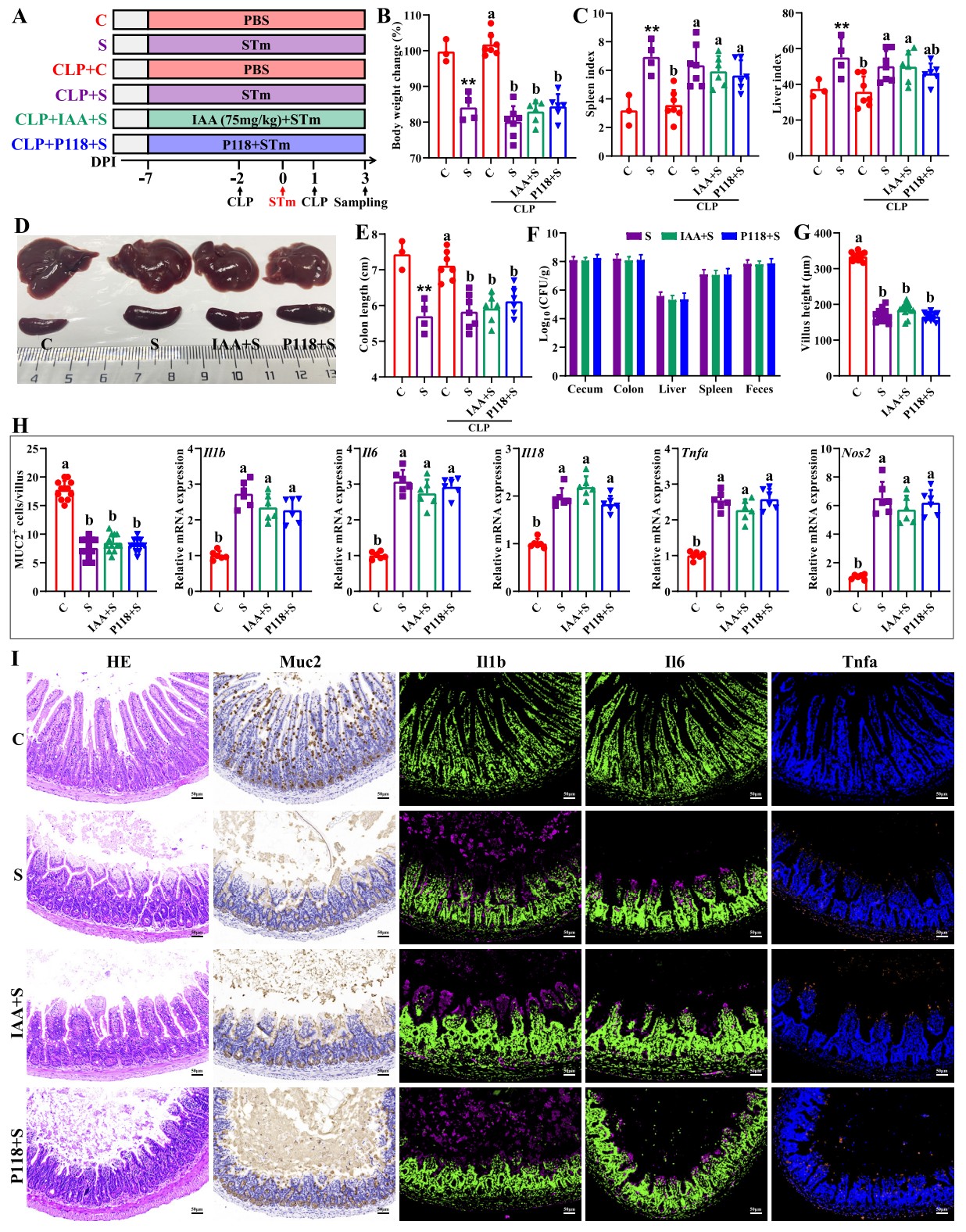

**Figure 6.** Macrophage depletion abrogates the protective effect of *L. rhamnosus* P118 and indole-3-acrylic acid against *S.* Typhimurium infection.
(**A**) Experimental design. (**B**) Body weight change. (**C**) Spleen and liver indexes. (**D**) Representative images of spleen and liver. (**E**) Colon length.
(**F**) *S.* Typhimurium burden in tissues and shedding in feces. (**G**) Villus height of ileum. (**H**) Muc2-positive cells and mRNA expression levels in ileum.
(**I**) Representative images of H&E staining and immunostaining in the ileum. Different lowercase letters indicate a significant difference (p<0.05).
(**B–E**) Significant differences **p<0.01 indicates 'S vs. C'. CLP: macrophage depletion reagent clodronate liposomes. C: PBS group; S: *S.* Typhimurium-infected group; IAA + S: indole-3-acrylic acid protective group; P+S: P118 protective group.

*Salmonella* infection-induced intestinal mucosa damage (*Figure 6G–I*) and pro-inflammatory responses (*Figure 6H and I*). Taken together, these findings indicate that P118 and IAA protect against *Salmonella* infection by inhibiting pro-inflammatory responses of intestinal macrophage.

## Discussion

Fermented foods and mammalian intestines are rich in lactic acid bacteria that can produce organic acids and lactobacillin to inhibit overgrowth, adhesion, invasion, and toxin secretion of pathogens (*Han et al., 2024*; *Yaqoob et al., 2024*). Traditionally, the culturomics strategy, together with multiple evaluation assays, is the primary way to search potential probiotic species from the complex microbial community, which is both time- and labor-consuming (*Wang et al., 2022*). Although the whole-genomic sequencing-based approach is an emerging favorable way to screen potential probiotic strains, it is hard to widely adopt based on numerous isolates due to the high sequencing costs (*Peng et al., 2023*). Given the similarities of gut morphology and physiological function with human and animal, *C. elegans* provides an invaluable model to investigate host-microbe interactions and probiotic properties of beneficial microbes at low cost (*Kumar et al., 2020*; *Poupet et al., 2020*). In the present study, we found *Salmonella* infection significantly induced death and decreased the lifespan of *C. elegans*, indicating *C. elegans* is very susceptible to *S.* Typhimurium infection, consistent with a previous report (*Rangan et al., 2016*). Integrating the results of culturomics and *C. elegans* infection strategies, *L. rhamnosus* P118 with excellent probiotic properties was screened from 290 isolates and exhibited broad in vitro antibacterial activities. Interestingly, high fermented dose and anaerobic cultures of P118 showed more antibacterial effects than low dose and aerobic cultures, indicating that the active factors were more released by P118 at the stationary phase under anaerobic conditions. What's more, the antibacterial activities of the fermented supernatant were sensitive to alkalinity, trypsin, proteinase-K, pepsin, and high temperature, indicating that the active secreted factors in the fermented supernatant were protein- or peptide-based active agents, consistent with a previous report that *L. rhamnosus* exerted antibacterial effects by secreting bioactive peptides (*Iram et al., 2022*). Although limited studies have reported bacteriocin-like activity in *L. rhamnosus*, it remains unclear whether this species produces active bacteriocins (*Kankainen et al., 2009*). Nonetheless, genomic data in this study revealed the presence of putative bacteriocin synthesis genes in the P118 genome. In agreement with a previous study (*Kankainen et al., 2009*), the predicted bacteriocins showed high homology with known class II bacteriocins (e.g., IIa carnobacteriocin B2 produced by *C. maltaromaticum*, IIb enterocin X produced by *E. faecium* KU-B5), which needs to be further validated by gene-editing technology and synthetic biology.

To further validate the antibacterial activities of *L. rhamnosus* P118 against enteric pathogens, *Salmonella*-infected mice model was employed. We found that administration of P118 significantly reduced mice susceptibility to *S.* Typhimurium infection by improving intestinal health, dysbiosis, and the changes of fecal metabolite profiles. Colonized *S.* Typhimurium in the intestine disrupts structural and functional intestinal integrity, leading to substantial immune response and severe epithelial barrier damage, and then migrates to internal organs (liver, spleen) via the blood circulatory system, causing splenomegaly and hepatomegaly (*Rogers et al., 2023*; *Wang et al., 2021a*). In the present study, P118 significantly increased *Salmonella*-infected mice survival and alleviated *Salmonella*-induced splenomegaly and hepatomegaly. The beneficial attenuated effect was partly contributed to by the reduced *Salmonella* colonization in the intestine and invasion into the peripheral organs, and eventually, the decreased fecal shedding, consistent with previous reports (*Liu et al., 2023*; *Wu et al., 2024*). It is reported that *S.* Typhimurium exploits and disrupts intestinal barrier integrity to adhere, and disseminate deeper into intestinal mucosa, and evade immune clearance (*Bernal-Bayard and Ramos-Morales, 2018*; *Hiyoshi et al., 2022*). In the current study, P118 treatment also significantly improved the structural and functional intestinal integrity, attenuated inflammatory responses, reduced the numbers of F4/80-positive macrophages and proinflammatory macrophages (F4/80$^+$ Nos2$^+$ cells), and increased the numbers of Ki67-positive and Muc2-positive goblet cells in the ileum, demonstrating that P118 improved the intestinal health and reduced pro-inflammatory responses of intestinal macrophages, which further confirms the beneficial alleviated effect exerted by *L. rhamnosus* P118.

Intestinal microbes play essential roles in providing colonization resistance against enteric pathogens and pathobionts and limiting adhesion, invasion, and transmission of pathogens (*Caballero-Flores et al., 2023*; *Jacobson et al., 2018*). Microbial dysbiosis and excessive immune responses

induced by enteric pathogens can, in turn, aggravate and exacerbate gut permeability and damage (*Read et al., 2021*; *Rogers et al., 2023*). Many studies have reported that *Salmonella* infection was often accompanied by microbial dysbiosis (*Gillis et al., 2018*; *Read et al., 2021*; *Wang et al., 2021a*). The current results found that *L. rhamnosus* P118 treatment significantly restored the bacterial community structures in *Salmonella*-infected mice and significantly reduced the relative abundances of potentially harmful microbes (e.g., *Salmonella*, *Klebsiella*, *Anaeroplasma*, *Morganella*), which belong to pathogens and pathobionts (*Li et al., 2025*; *Galán, 2021*; *Laupland et al., 2022*). Structure composition and assembly of microbial communities are essential for ecosystem function (*Shi et al., 2023*; *Xun et al., 2019*), and revealing the underlying mechanisms of microbial community assembly is a major goal of microbial community ecology (*Jiao et al., 2020*; *Stegen et al., 2012*). Generally, stochastic (e.g., dispersal events, ecological drift, random birth, death) and deterministic (e.g., interspecies interactions [e.g., competition, facilitation, mutualisms, and predation], species traits, environmental factors) processes are two major ecological processes that drive microbiome assembly (*Stegen et al., 2012*; *Zhou and Ning, 2017*). Deterministic processes mainly involve nonrandom and niche-based mechanisms with abiotic and biotic factors that influence microbial community assembly (*Vellend, 2010*), whereas stochastic processes mainly reflect random changes in the relative abundance of species (*Jiao et al., 2020*; *Jiao et al., 2021*). Although important in shaping the diversities of microbial composition and functions, the relative contribution of microbiome assembly processes varies with different habitats (*Jiao et al., 2020*; *Jiao et al., 2021*). The present study revealed that the stochastic processes played dominant roles in intestinal microbiome assembly in *Salmonella*-infected mice, indicating random changes occurred in the relative abundance of species. Conversely, the deterministic processes played more significant roles in driving intestinal microbial community assembly in P118-pretreated mice than in *Salmonella*-infected mice, indicating abiotic factors (e.g., dietary intervention) or biotic interactions (e.g., microbial competition, facilitation, mutualism, predation, host filtering) might influence the microbial community.

Accumulating evidence revealed that gut microbes-derived metabolites (e.g., short-chain fatty acids, tryptophan/indole-derived metabolites, bacteriocins, bile acid, natural products) are crucial mediators in host physiological activities (*Cani, 2019*; *Wlodarska et al., 2017*), and metabolic disorders are major risk factors for bacterial infections (*Olive and Sassetti, 2016*). In this study, the results showed that *L. rhamnosus* P118 improved *Salmonella* infection-induced fecal metabolite changes. Interestingly, P118 significantly increased the fecal levels of tryptophan and its derivatives (indole, IAA, 5-hydroxytryptophan), and those metabolites were negatively correlated with *S.* Typhimurium burdens (in the duodenum, colon, liver, spleen, feces) and organ (spleen, liver) indices, and positively correlated with the body weight of mice, indicating that microbiota-derived tryptophan/indole metabolites play beneficial roles in P118-mediated probiotic activities in enhancing host tolerance to *Salmonella* infection. It is reported that through binding to the AHR of host, bacterial tryptophan derivatives act as triggers to stimulate immune responses and gut hormones to exert anti-inflammatory activities, enhance intestinal epithelial barrier, and promote gastrointestinal motility, which contributes to host gastrointestinal homeostasis and health (*Agus et al., 2018*; *Cani, 2019*). Our genomic and metabolomic data further revealed that P118 could produce a wide array of tryptophan-derived metabolites owing to encoding a variety of enzyme genes necessary to metabolize tryptophan into indole derivatives; consistently, some *Lactobacillus* species broadly conserved enzyme encoding genes that are involved in tryptophan metabolism (*Montgomery et al., 2022*). The above results indicate that *L. rhamnosus* P118- and microbe-derived tryptophan/indole metabolites might play positive roles in P118-mediated probiotic activities. Although significant increases in microbe-derived tryptophan/indole metabolites were observed in P118-treated mice, and it was established that P118 can metabolize tryptophan into indole derivatives, it remains to be investigated whether the differential tryptophan/indole metabolites were directly derived from P118 or from other microbes. To test the roles of differential tryptophan/indole metabolites in P118-mediated beneficial effects, exogenous IAA, which significantly and negatively correlated with *S.* Typhimurium burdens and hepatosplenomegaly mentioned above, was employed to investigate its protective effect against *Salmonella* infection. Interestingly, we found that IAA administration significantly enhanced mice tolerance to *S.* Typhimurium infection by improving intestinal health and attenuating pro-inflammatory responses of intestinal macrophages. Furthermore, the in vitro results also provide that IAA enhanced bactericidal capacity of RAW 264.7 cells and inhibited *S.* Typhimurium-induced pro-inflammatory responses, consistent with

a previous report (*Wlodarska et al., 2017*). Taken together, these results demonstrate that *L. rhamnosus* P118/microbe-derived IAA plays beneficial roles in P118-mediated probiotic activities.

Macrophages are important in initiating immune responses, maintaining intestinal immune homeostasis, phagocytic clearance of pathogens, tissue repair, and host defense (*Na et al., 2019*; *Neurath, 2024*). It is reported that macrophages with strong plasticity could be polarized into pro-inflammatory (M1) phenotype or proresolving (M2) phenotype (*Na et al., 2019*). M1 macrophage activation is crucial and necessary for pathogen clearance (*Na et al., 2019*). However, exaggerated immune responses triggered by M1 macrophages are detrimental to the host by inducing tissue damage and cell deaths (*Wang et al., 2020*). The current study showed that both *L. rhamnosus* P118 and IAA significantly attenuated *S.* Typhimurium infection-induced pro-inflammatory responses, indicating that pro-inflammatory macrophages might be key targets for *L. rhamnosus* P118 exerting probiotic effect. This hypothesis was confirmed by macrophage depletion experiments that the protective effects of *L. rhamnosus* P118 and IAA against *Salmonella* infection were abrogated after intestinal macrophage depletion. Taken together, these results indicate that *L. rhamnosus* P118/microbe-derived indole metabolites enhance host tolerance to *Salmonella* infection by reducing intestinal pro-inflammatory responses, which also provides a potential alternative strategy for using probiotics and their derived metabolites to treat intestinal inflammatory disorders such as inflammatory bowel disease (*Lavelle and Sokol, 2020*; *Neurath, 2024*).

## Conclusion

In summary, two distinctive approaches were used to screen the probiotic candidate, and the P118 strain was underscored for beneficial effects on a murine infection model. Further, bacterial genomic sequencing, gut microbiota, and metabolomic analysis pinpointed the microbe-derived tryptophan/indole could be the importance of P118 probiotic properties. Nevertheless, the newly found P118 could enhance host tolerance to *Salmonella* infections via various pathways, including direct antibacterial actions, inhibiting *Salmonella* colonization and invasion, attenuating pro-inflammatory responses of intestinal macrophages, and modulating gut microbiota mediated by microbe-derived indole metabolites (*Figure 7*). Further investigations are needed to assess whether the mechanisms observed in P118 are strain-specific or broadly applicable to other *L. rhamnosus* strains, or LAB species in general.

## Materials and methods
### Bacterial isolation and culture

Lactic acid bacteria (LAB) and *Enterococcus* strains were isolated from 39 samples: 33 fermented yogurt samples (collected from families in multiple cities of China, including Lanzhou, Urumqi, Guangzhou, Shenzhen, Shanghai, Hohhot, Nanjing, Yangling, Dali, Zhengzhou, Shangqiu, Harbin, Kunming, Puer) and 6 healthy piglet rectal content samples without pathogen infection and diarrhea in pig farm of Zhejiang province (*Table 1*). Ten isolates were randomly selected from each sample. De Man-Rogosa-Sharpe (MRS) with 2.0% $CaCO_3$ and brain heart infusion (BHI) broths (Huankai Microbial, Guangzhou, China) were used for bacteria isolation and cultivation. MALDI-TOF MS (Bruker Daltonik GmbH, Bremen, Germany) method was employed to identify bacterial species with a confidence level ≥90% (*He et al., 2022*). *L. rhamnosus* P118 (P118) and *Salmonella enterica* serovar Typhimurium SL1344 (STm) were cultured in MRS and Luria Broth (LB) medium at 37°C overnight under aerobic conditions, separately. The final concentration of bacterial isolates was constantly checked by the spreading plate method (*Wang et al., 2023*). After being centrifuged at 4000×*g* for 15 min at 4°C, the fermented supernatant of *L. rhamnosus* P118 was collected by centrifugation at 4000×*g* for 15 min at 4°C, and then filtered through a 0.22 µm membrane (Merck Millipore, Burlington, MA, USA) kept at 4°C for further use.

### Probiotic property evaluation of candidate isolates

For milk-clotting activity analysis of probiotics (*Zhang et al., 2023*), overnight fermented lactic acid bacteria isolates under logarithmic growth were added to test tubes containing skim milk (10% skim milk with 10 mM $CaCl_2$) at 1:20 ratio (v/v) and then incubated at 37°C. The coagulating time, final pH value, and acidity of milk were measured during the curdling process.

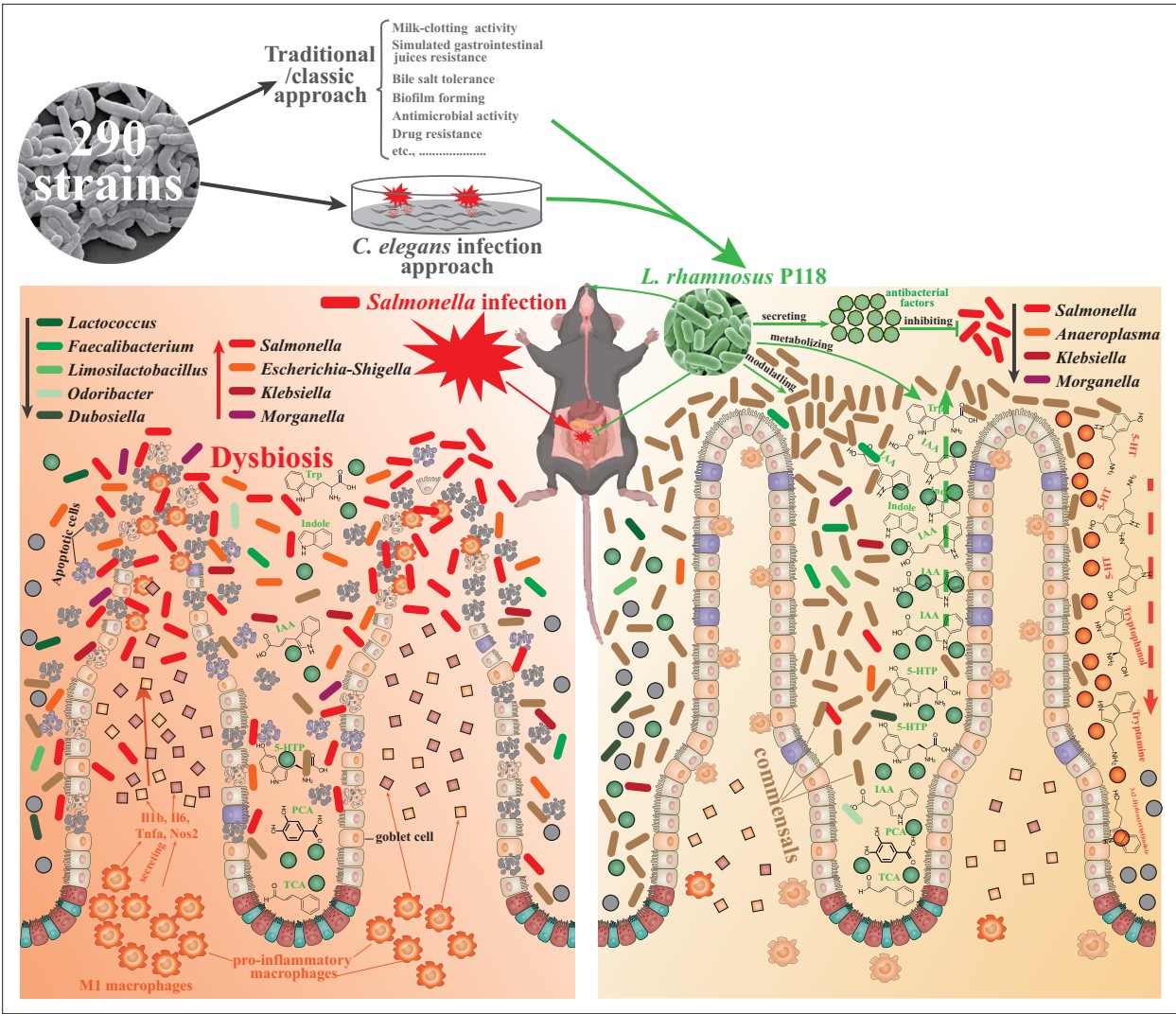

**Figure 7.** *L. rhamnosus* P118 strain with great probiotic properties was screened from 290 purified isolates through two distinctive screen approaches, and P118 strain was underscored for protective effects on a murine infection model. Further, multi-omics analysis pinpointed the microbe-derived tryptophan/indole could be the importance of P118 probiotic properties. Nevertheless, the newly found P118 enhances host tolerance to *Salmonella* infections via various pathways, including direct antibacterial actions, inhibiting *Salmonella* colonization and invasion, attenuating pro-inflammatory responses of intestinal macrophages, and modulating gut microbiota mediated by microbe-derived indole metabolites.

The bile salt tolerance assay was conducted according to a previous report (*Wang et al., 2023*). Briefly, the isolates under logarithmic growth were resuspended into fresh MRS medium (1% v/v) containing different (wt/v, 0, 0.3%, 0.6%, 1.0%, 1.5%, 2.0%) porcine bile (MACKLIN, Shanghai, China) and then plated into 96-well plates. After incubation for 24 h at 37°C, the absorbance at 620 nm was measured by a SpectraMax M5 reader (Molecular Devices, Sunnyvale, CA, USA).

Biofilm-forming assay of the isolates was performed as described previously (*Zhou et al., 2023*). Briefly, 200 μL overnight cultures (adjusted to 2×10⁶ CFU/mL) were added into 96-well flat-bottom plates. After incubation for 36 h at 37°C, the planktonic cells were discarded. After being gently washed three times with sterile PBS, the adherent cells were stained with 200 μL of 0.4% crystal violet for 15 min at room temperature and then washed with sterile PBS three times again. Finally, 200 μL 75% ethanol was added to dissolve the stained adherent cells for 15 min, and biofilm was quantified by measuring the absorbance at 590 nm using a SpectraMax M5 reader.

For antimicrobial susceptibility test, 12 antimicrobial agents (penicillins, erythromycin, lincomycin, gentamicin, doxycycline, ceftriaxone, norfloxacin, azithromycin, fleroxacin, vancomycin, streptomycin)

were employed to evaluate the antimicrobial susceptibility of the candidate probiotics by utilizing the Kirby-Bauer disk diffusion method (*Hudzicki, 2009*).

The agar diffusion method (*Wang et al., 2023*) was used to evaluate the antimicrobial activity against multiple pathogens in vitro. Briefly, bacterial culture medium (LB, BHI) containing 0.25% (v/v) different zoonotic pathogens under logarithmic growth was poured into 10 cm plates (Corning, NY, USA), separately. Then, 200 μL of the fermented sterile supernatant collected from the overnight cultured *L. rhamnosus* P118 was added into 8 mm agar wells created by Oxford cup. The bacterio-static effect was evaluated by the growth inhibition zone around the 8 mm agar wells after overnight incubation at 37°C.

### *Caenorhabditis elegans* infection

*C. elegans* infection model was performed as previously described (*Rangan et al., 2016*). *C. elegans* N2 (Bristol) were routinely raised on Nematode Growth Media (NGM) feeding with *E. coli* OP50, and L4-stage worms were prepared according to the description in WormBook (*Girard et al., 2007*). 200 μL of the isolated isolates (adjusted to $5\times10^8$ CFU/mL using M9 medium) and *L. rhamnosus* P118-related substances (e.g., mixed cultures, fermented sterile supernatants, P118 lysate, dead P118) were poured onto NGM in 24-well plates, separately, and dried at 22°C for 4 h. L4-stage worms were then transferred to the lawns of the poured NGM plates (30 worms/plate) with three plates per group for 24 h. Then, the treated worms were washed and transferred to lawns of STm ($1\times10^8$ CFU/lawn) on NGM plates for infection. Worm survival was monitored at 24 h intervals for 16 days.

### Whole-genome sequencing and data analysis

Whole-genome sequencing, quality control, assembly, and annotation of *L. rhamnosus* P118 were conducted according to our previous study (*Zhou et al., 2023*) using PacBio Sequel platform and Illumina NovaSeq PE150 at Novogene (Novogene Co, Ltd., Beijing, China), and the P118 draft genome is available in the Sequence Read Archive under accession number PRJNA848987. *L. rhamnosus* P118 assembly was queried against the NCBI non-redundant prokaryotic genomes database using the Microbial Genomes Atlas (MIGA) webserver (database update to 10/10/2023) (*Rodriguez-R et al., 2018*). Taxonomic classification was inferred by the maximum average amino acid identity (AAI) against all genomes in the database, with p-values estimated from the distribution of all the reference genomes in NCBI's RefSeq at each taxonomic level as a readout of classification proba-bility. Average nucleotide identity (ANI) and AAI tables of maximally the top 50 reference hits in the database were extracted and graphed as x-y scatter plots to determine the nearest phylogenetic neighbors. To generate phylogenic trees of isolates and MIGA-identified nearest phylogenetic neigh-bors, the interfered proteome from PROKKA isolate annotations, or as publicly available in NCBI for applicable reference genomes, was uploaded to the ANI/AAI-Matrix calculator (*Rodriguez-R and Konstantinidis, 2016*). The resulting phylogenetic tree based on AAI was visualized using the interac-tive tree of life (iTOL) (*Letunic and Bork, 2021*).

*L. rhamnosus* P118 tryptophan-associated enzymes were identified as previously described (including TNA, TMO, TrpD, ArAT, ALD, IPDC, FldH, and AmiE) (*Montgomery et al., 2022*). Briefly, enzyme commission numbers, where available, or alternatively enzyme names, were queried in PATRIC, PATRIC Global Family (PGF) cross-genus identifiers were extracted and compiled for query within the *L. rhamnosus* P118 genome. Protein sequences of *L. rhamnosus* P118 identified enzymes were analyzed using InterProScan for additional functional prediction, and sequence homology to the previously experimentally validated enzymes within other bacterial species was determined using Blastp at 30% cross-genus identity and 90% identity within each species.

### *S.* Typhimurium murine infection model

Animal assays, specifically 80 six-week-old female C57BL/6 mice (Slac Animal Inc, Shanghai, China), were conducted in the Laboratory Animal Center of Zhejiang University under light-controlled (12 h light/dark cycle), temperature-controlled (22 ± 2°C), and humidity-controlled (55 ± 5°C) conditions. Forty mice were randomly selected for survival evaluation under *S.* Typhimurium infection, and the rest were used for sample collection on day 3 post-infection (*Figure 2A*). The mice were randomly divided into four groups (n=10/group): control group (C), *L. rhamnosus* P118 group (P), *S.* Typhimurium-infected group (S), and *L. rhamnosus* P118 protective group (P+S). Mice in the C and S groups were

drinking sterile water every day, respectively. Mice in the P and P+ST groups were orally administered *L. rhamnosus* P118 (1×10⁸ CFU/mouse) daily for 7 days. Then, mice were orally ingested with *S.* Typhimurium SL1344 (1×10⁸ CFU/mouse) or sterile PBS. All mice were allowed free access to water and food, and weighed every day. Mice were euthanized on day 3 post-infection. Spleen and liver were weighed for organ index calculation according to the formula: organ index = organ weight (g)/body weight (g) * 1000.

To evaluate the protective effect of IAA, 18 mice were randomly divided into three groups (n=6/group): control group (C), *S.* Typhimurium-infected group (S), and IAA protective group (IAA + S). Mice in the C and S groups were drinking sterile water every day, respectively. IAA (Sigma, I3807) was dissolved in 1 M NaOH in PBS and adjusted pH to 7.4 with 1 M HCl. Mice in the IAA + ST groups were orally administered with 75 mg/kg IAA daily for 7 days. Then, mice were orally ingested with *S.* Typhimurium SL1344 or sterile PBS. At 3 days post-infection, mice were euthanized for sample collection.

### In vivo macrophage depletion

To evaluate the involved roles of macrophages, macrophage depletion assay was conducted through intraperitoneal injection with 250 µL/mouse clodronate liposomes (CLP, From Vrije Universiteit Amsterdam) twice (2 days prior and 1 day after *Salmonella* infection, separately) according to the manufacturer's instructions. Thirty-three mice were randomly divided into six groups (n=3–6/group): control group (C), *S.* Typhimurium-infected group (S), control group with macrophage depletion (CLP + C), *S.* Typhimurium-infected group with macrophage depletion (CLP + S), IAA protective group with macrophage depletion (CLP + IAA + S), and P118 protective group with macrophage depletion (CLP + P + S). At 3 days post-infection, mice were euthanized for sample collection.

### Macrophage cell culture

Murine macrophage cell line RAW 264.7 purchased from ATCC was cultured in DMEM/F12 medium (Gibco, Carlsbad, CA) supplemented with 10% FBS (Gibco) and 1% (v/v) antibiotic solution (100 µg/mL streptomycin + 100 U/mL penicillin, Sigma-Aldrich, St. Louis, MO) at 37°C in a 5% humidified $CO_2$ incubator. If not mentioned, the antibiotic solution (100 µg/mL streptomycin and 100 U/mL penicillin) was not added into DMEM/F12 medium in the further experiment.

For *S.* Typhimurium killing assay, RAW 264.7 cells seeded into 12-well plates (2×10⁶ cells/well) were treated with 100 µM IAA for 6 h, followed by infecting with *S.* Typhimurium (MOI = 10) for 1 h. After washing three times with sterile PBS, the infected RAW 264.7 cells were incubated for 0 h, 6 h, and 12 h in DMEM/F12 medium containing gentamicin (50 µg/mL). At each time point, the infected RAW 264.7 cells were washed with sterile PBS four times and lysed with 0.01% Triton X-100 diluted in PBS. The serial 10-fold dilutions of cell lysates were immediately plated on LB agar plates to determine bacterial viability.

For the immune response assay, RAW 264.7 cells seeded into 12-well plates (2×10⁶ cells/well) were pretreated with CH-223191 (AHR inhibitor, 10 µM) for 6 h, subsequently treating cells with 100 µM IAA for another 6 h. After 6 h, RAW 264.7 cells were infected with *S.* Typhimurium (MOI = 10) for 2 h. After washing three times with sterile PBS, the infected RAW 264.7 cells were collected using RNAiso Plus (TaKaRa, Dalian, China) for total RNA extraction and quantitative real-time PCR (qPCR). The primer sets are listed in **Supplementary file 6**.

### *S.* Typhimurium burden

*S.* Typhimurium loads in tissues (duodenum, cecum, colon, liver, and spleen) and shedding in feces were determined as previously described (*Xu et al., 2018*). Briefly, tissues and fecal samples were collected under sterile conditions and homogenized in sterile PBS containing 0.1% Triton X-100. Serial 10-fold dilutions of tissue homogenates were spread onto SS agar plates in triplicate and then incubated at 37°C overnight to quantify the bacterial colony-forming unit (CFU).

### Histopathology, immunofluorescent analysis, transmission electron microscope (TEM) observation, and clinical symptom score

Tissues (ileum, liver, and spleen) fixed in 4% paraformaldehyde were embedded in paraffin, sliced, dehydrated, and then sectioned for hematoxylin and eosin (H&E) staining. The tissue slices were imaged and analyzed by using Olympus Microscope (Tokyo, Japan). For the immunofluorescent

assay, the paraffin-embedded ileum samples were incubated with primary antibodies against Ki67 (GB121141, Servicebio, China), lysozyme (GB11345, Servicebio), Muc2 (GB11344, Servicebio), F4/80 (GB113373, Servicebio), Nos2 (GB11119, Servicebio), Il1b (GB11113, Servicebio), Il6 (GB11117, Servicebio), Tnfa (GB11188, Servicebio) overnight at 4°C and then further incubated with secondary antibodies (Alexa Fluor 488-conjugated goat anti-rabbit/mouse IgG) or HRP-conjugated goat anti-rabbit secondary antibody (GB23303, Servicebio) for 60 min at room temperature. Finally, ileum slices were digitalized by Pannoramic MIDI (3DHISTECH, Hungary).

TEM observations of ileum tissues were prepared according to our previous study (*Peng et al., 2022*). Briefly, after being fixed with 2.5% buffered glutaraldehyde, ileum segments were washed three times by cold 100 mM phosphate buffers, and then post-fixed in 0.1% osmium tetroxide for 2 h, rapidly dehydrated in ascending grades of ethanol (30, 50, 70, 95, and 100%), and moved into a 1:1 mixture of propylene oxide and epoxy araldite. Finally, the ileum samples were observed and captured by the transmission electron microscope (JEOL, Tokyo, Japan).

The clinical symptom scores were evaluated and graded by two blinded assessors according to previous reports with modified (*Burkholder et al., 2012*), and the scoring criteria are listed in *Supplementary file 7*.

## ELISA assay and qPCR

The serum contents of interleukin (IL)-1β and IL-18 were determined by ELISA kits (Solarbio, Beijing, China) according to the manufacturer's instructions.

Total RNA extracted from intestinal tissues and RAW 264.7 cells by RNAiso Plus kit (TaKaRa) was reverse-transcribed using PrimeScript II 1st Strand cDNA Synthesis Kit (TaKaRa) according to the manufacturer's instructions. The qPCR was then performed on the StepOne real-time PCR system (Applied Biosystems) using SYBR PremixEx TaqII (TaKaRa). The primer sets are listed in *Supplementary file 6*. Fold changes were calculated after normalizing to the housekeeping gene *Gapdh* using the $2^{-\Delta\Delta Ct}$ method (*Livak and Schmittgen, 2001*).

## Microbiota analysis

The TIANamp Stool DNA Kit (Tiangen, Beijing, China) was employed to extract fecal bacterial genomic DNA, and fecal bacterial communities were investigated using 16S rDNA sequencing by targeting the V3-V4 hypervariable region. The 16S rDNA sequencing was then performed on an Illumina NovaSeq platform (Illumina Inc, San Diego, CA, USA). The quality filter of the paired-end raw sequences and a cluster of the filtered sequences into the OTU at 97% similarity was performed by QIIME software (version 1.9.1). Microbial OTU representative sequences were assigned to a taxonomic lineage by the RDP classifier based on the SILVA database (version 132 release).

To investigate the dissimilarities in bacterial communities and fecal metabolomes, principal coordinates analysis (PCoA), analysis of similarities (ANOSIM), permutational multivariate analysis of variance (PERMANOVA), and multi-response permutation procedure (MRPP) were calculated using 'vegan' package and visualized using 'ggplot2' package. Difference analysis of bacterial communities based on OUT levels among groups was calculated using the 'DESeq2' package and was envisioned by the 'UpSetR' and 'ggplot2' packages. The significant differences in bacterial taxonomies were analyzed and visualized by statistical analysis of taxonomic and functional profiles (STAMP) software with a two-sided Welch's *t*-test (*Parks et al., 2014*).

The Sloan neutral model was employed to estimate the importance of neutral processes in the assembly of bacterial communities using the 'MicEco' package (*Sloan et al., 2006*). A null-model-based statistical framework was conducted to evaluate the relative importance of determinism and stochasticity in bacterial community assembly (*Stegen et al., 2013*; *Xu et al., 2022*). Two assembly processes were defined as deterministic processes (|βNTI|>2) and stochastic processes (|βNTI|≤2) based on the absolute values of beta Nearest Taxon Index (βNTI) (*Jiao et al., 2020*). Additionally, five ecological assembly processes were then interpreted as homogeneous selection (βNTI<-2), homogenizing dispersal (|βNTI|≤2 and $RC_{bray}$ <−0.95), undominated (|βNTI|≤2 and |$RC_{bray}$|<0.95), dispersal limitation (|βNTI|≤2 and $RC_{bray}$ >0.95) and variable (heterogeneous) selection (βNTI >2) based on the threshold of the absolute values of βNTI and Bray–Curtis-based Raup–Crick ($RC_{bray}$) (*Jiao et al., 2020*; *Xu et al., 2022*).

## Metabolomics analysis

Untargeted metabolomics was investigated to analyze the fecal metabolomes of mice. The method for extracting metabolites from feces was conducted according to our previous study described (*Wang et al., 2021b*). After extracting metabolites from feces, UHPLC-MS/MS analyses of samples were performed by Vanquish UHPLC systems (Thermo Fisher, Germany) coupled with Orbitrap Q Exactive HF-X mass spectrometers (Thermo Fisher) in Novogene Co., Ltd. (Beijing, China). Compound Discoverer 3.1 (CD3.1, Thermo Fisher) was employed to conduct peak alignment and quantify metabolites generated by UHPLC-MS/MS. After normalizing to total spectral intensity and predicting molecular formula, the peaks were queried against mzCloud, mzVault, and MassList databases to extract the quantitative results. Annotation of fecal metabolite signatures was queried against the KEGG database, HMDB database, and LIPIDMaps database. Significant differences in fecal metabolites were analyzed using univariate analysis (*t*-test) and partial least-squares discriminant analysis (PLS-DA), in which fecal metabolites with variable importance in the projection (VIP) >1 and $p < 0.05$ were considered differential metabolites. Pearson correlation analysis between *S*. Typhimurium burden (in tissues and feces) and fecal metabolites was analyzed and visualized using the 'linkET' package (*Huang, 2021*). MetOrigin platform was employed to analyze the sources of microbial metabolites and their metabolic functions in fecal metabolomes (*Yu et al., 2022*).

## Statistical analysis

The significance of the results was analyzed by one-way analysis of variance (ANOVA) with Tukey's multiple comparisons test using SPSS v24 (SPSS Inc, Chicago, IL, USA), and statistical graphs were visualized using GraphPad Prism v8.0 (GraphPad Software, CA) and package 'ggplot2' of R software (v4.3.1). * $p < 0.05$, ** $p < 0.01$, *** $p < 0.001$.

## Acknowledgements

This work was supported by the National Program on Key Research Project of China (2022YFC2604201) as well as the European Union's Horizon 2020 Research and Innovation Programme under Grant Agreement No. 861917-SAFFI, Zhejiang Provincial Key R&D Program of China (2023C03045, 2022C02024), China Postdoctoral Science Foundation under Grant Number 2024M750693, Key Research and Development Program of Hangzhou (202203A08), and Zhejiang Provincial Natural Science Foundation of China (LQ23C180005).

## Additional information

### Funding

| Funder | Grant reference number | Author |
|---|---|---|
| National Program on Key Research Project of China | 2022YFC2604201 | Min Yue |
| European Union's Horizon 2020 Research and Innovation Programme | 861917-SAFFI | Min Yue |
| Zhejiang Provincial Key R&D Program of China | 2023C03045 | Min Yue |
| Zhejiang Provincial Key R&D Program of China | 2022C02024 | Min Yue |
| China Postdoctoral Science Foundation | 2024M750693 | Baikui Wang |
| Key Research and Development Program of Hangzhou | 202203A08 | Min Yue |

| Funder | Grant reference number | Author |
|---|---|---|
| Zhejiang Province Natural Science Foundation of China | LQ23C180005 | Min Yue |

The funders had no role in study design, data collection and interpretation, or the decision to submit the work for publication.

## Author contributions

Baikui Wang, Conceptualization, Software, Formal analysis, Funding acquisition, Validation, Investigation, Visualization, Methodology, Writing – original draft, Writing – review and editing; Xianqi Peng, Conceptualization, Validation, Investigation, Visualization, Methodology, Writing – original draft, Writing – review and editing; Xiao Zhou, Jiayun Yao, Haiqi Zhang, Investigation, Methodology; Xiuyan Jin, Investigation, Visualization, Methodology; Abubakar Siddique, Investigation, Visualization; Weifen Li, Investigation, Writing – review and editing; Yan Li, Supervision, Writing – review and editing; Min Yue, Conceptualization, Resources, Data curation, Supervision, Funding acquisition, Validation, Investigation, Methodology, Project administration, Writing – review and editing

## Author ORCIDs

Baikui Wang (iD) https://orcid.org/0000-0001-5283-7337
Xiao Zhou (iD) https://orcid.org/0000-0001-6510-4095
Yan Li (iD) https://orcid.org/0000-0003-4813-5783
Min Yue (iD) https://orcid.org/0000-0002-6787-0794

## Ethics

Animal procedures followed the guidelines and were approved by the Institutional Animal Care and Use Committee of Zhejiang University (Permission number: ZJU20220295).

Reviewer #1 (Public review): https://doi.org/10.7554/eLife.101198.4.sa1
Reviewer #2 (Public review): https://doi.org/10.7554/eLife.101198.4.sa2
Author response https://doi.org/10.7554/eLife.101198.4.sa3

---

# Additional files

## Supplementary files

Supplementary file 1. Milk-clotting activity of lactic acid bacteria.

Supplementary file 2. Significance between life span data was calculated using a Mantel–Cox log-rank test.

Supplementary file 3. Significance between life span data was calculated using a Mantel–Cox log-rank test.

Supplementary file 4. Species-level identity of *L. rhamnosus* P118. The draft genome of P118 was queried using the NCBI non-redundant prokaryotic database available through MIGA (http://microbial-genomes.org/). p-Values reflect confidence in the assignment as each taxonomic rank.

Supplementary file 5. *L. rhamnosus* P118 draft genome nearest subspecies phylogenetic neighbors. Average nucleotide identity (ANI) and average amino acid identity (AAI) using both percent identity and a fraction of genome shared as determined through MIGA (http://microbial-genomes.org/) query of the NCBI non-redundant prokaryotic genomes database were considered for ranking likelihood of nearest subspecies phylogenetic neighbors for duplicate isolates of each *Lacticaseibacillus* species.

Supplementary file 6. Primers used for the quantitative PCR.

Supplementary file 7. The criteria of clinical symptom scores.

Supplementary file 8. ANOSIM, PERMANOVA, and MRPP analysis of microbial diversity among treatments.

Supplementary file 9. ANOSIM, PERMANOVA, and MRPP analysis of fecal metabolites among groups.

Supplementary file 10. Significant differential fecal metabolites in 'S vs. C' and 'P+S vs. S'.

Supplementary file 11. Prediction of aminotransferase class and tryptophan binding affinity in *L. rhamnosus* P118.

Supplementary file 12. Tryptophan metabolism and indole derivatives in *L. rhamnosus* P118 cultures.

MDAR checklist

## Data availability

16S rRNA sequencing data is deposited at the Genome Sequence Archive (GSA) of the BIG Data Center (https://bigd.big.ac.cn/gsa/) under accession no. PRJCA021751. L. rhamnosus P118 isolate draft genome assembly is publicly available in the NCBI under accession no. PRJNA848987.

The following dataset was generated:

| Author(s) | Year | Dataset title | Dataset URL | Database and Identifier |
|---|---|---|---|---|
| Wang B | 2023 | Lacticaseibacillus rhamnosus P118-derived tryptophan/indole metabolites enhance host tolerance to Salmonella infection | https://ngdc.cncb.ac.cn/bioproject/browse/PRJCA021751 | Genome Sequence Archive, PRJCA021751 |

The following previously published dataset was used:

| Author(s) | Year | Dataset title | Dataset URL | Database and Identifier |
|---|---|---|---|---|
| Peng X | 2022 | Lacticaseibacillus rhamnosus strain:P118 Genome sequencing | https://www.ncbi.nlm.nih.gov/bioproject/PRJNA848987 | NCBI BioProject, PRJNA848987 |

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
