## [Editor Report · eLife Assessment]

The microbiome field is constantly providing insight on various health-related properties elicited by the commensals that inhabit their mammalian hosts. Harnessing the potential of these commensals for knowledge about host–microbe interactions, as well as properties with therapeutic implications, will likely remain a fruitful field for decades to come. In this **valuable** study, Wang et al. use various methods, encompassing classic microbiology, genomics, chemical biology, and immunology, to identify a potent probiotic strain that protects nematode and murine hosts from *Salmonella enterica* infection. The authors provide **compelling** evidence identifying gut metabolites that are correlated with protection and show that a single metabolite can recapitulate the effects of probiotic administration.

---

## [Referee Report · Reviewer #1 (Public review)]

Summary:

Diarrheal diseases represent an important public health issue. Among the many pathogens that contribute to this problem, *Salmonella enterica* serovar Typhimurium is an important one. Due to the rise in antimicrobial resistance and the problems associated with widespread antibiotic use, the discovery and development of new strategies to combat bacterial infections is urgently needed. The microbiome field is constantly providing us with various health-related properties elicited by the commensals that inhabit their mammalian hosts. Harnessing the potential of these commensals for knowledge about host-microbe interactions as well as useful properties with therapeutic implications will likely to remain a fruitful field for decades to come. In this manuscript, Wang et al use various methods, encompassing classic microbiology, genomics, chemical biology, and immunology, to identify a potent probiotic strain that protects nematode and murine hosts from *S. enterica* infection. Additionally, authors identify gut metabolites that are correlated with protection, and show that a single metabolite can recapitulate the effects of probiotic administration.

Strengths:

The utilization of varied methods by the authors, together with the impressive amount of data generated, to support the claims and conclusions made in the manuscript is a major strength of the work. Also, the ability the move beyond simple identification of the active probiotic, also identifying compounds that are at least partially responsible for the protective effects, is commendable.

Weaknesses:

No major weaknesses noted.

---

## [Referee Report · Reviewer #2 (Public review)]

Summary:

In this work, the investigators isolated one Lacticaseibacillus rhamnosus strain (P118), and determined this strain worked well against Salmonella Typhimurium infection. Then, further studies were performed to identify the mechanism of bacterial resistance, and a list of confirmatory assays were carried out to test the hypothesis.

Strengths:

The authors provided details regarding all assays performed in this work, and this reviewer trusted that the conclusion in this manuscript is solid. I appreciate the efforts of the authors to perform different types of in vivo and in vitro studies to confirm the hypothesis.

---

## [Author Response]

The following is the authors’ response to the previous reviews

**Public Reviews:**

**Reviewer #1 (Public review):**
Summary:Diarrheal diseases represent an important public health issue. Among the many pathogens that contribute to this problem, *Salmonella enterica* serovar Typhimurium is an important one. Due to the rise in antimicrobial resistance and the problems associated with widespread antibiotic use, the discovery and development of new strategies to combat bacterial infections is urgently needed. The microbiome field is constantly providing us with various health-related properties elicited by the commensals that inhabit their mammalian hosts. Harnessing the potential of these commensals for knowledge about host-microbe interactions as well as useful properties with therapeutic implications will likely to remain a fruitful field for decades to come. In this manuscript, Wang et al use various methods, encompassing classic microbiology, genomics, chemical biology, and immunology, to identify a potent probiotic strain that protects nematode and murine hosts from *S. enterica* infection. Additionally, authors identify gut metabolites that are correlated with protection, and show that a single metabolite can recapitulate the effects of probiotic administration.

We gratefully appreciate your positive and professional comments.

Strengths:The utilization of varied methods by the authors, together with the impressive amount of data generated, to support the claims and conclusions made in the manuscript is a major strength of the work. Also, the ability the move beyond simple identification of the active probiotic, also identifying compounds that are at least partially responsible for the protective effects, is commendable.

We gratefully appreciate your positive and professional comments.

Weaknesses:No major weaknesses noted.

We gratefully appreciate your positive comments.

**Reviewer #2 (Public review):**
Summary:In this work, the investigators isolated one Lacticaseibacillus rhamnosus strain (P118), and determined this strain worked well against Salmonella Typhimurium infection. Then, further studies were performed to identify the mechanism of bacterial resistance, and a list of confirmatory assays were carried out to test the hypothesis.

We gratefully appreciate your positive and professional comments.

Strengths:The authors provided details regarding all assays performed in this work, and this reviewer trusted that the conclusion in this manuscript is solid. I appreciate the efforts of the authors to perform different types of in vivo and in vitro studies to confirm the hypothesis.

We gratefully appreciate your positive and professional comments.

Weaknesses:I have mainly two questions for this work.Main point-1:The authors provided the below information about the sources from which Lacticaseibacillus rhamnosus was isolated. More details are needed. What are the criteria to choose these samples? Where were these samples originate from? How many strains of bacteria were obtained from which types of samples?Lines 486-488: Lactic acid bacteria (LAB) and Enterococcus strains were isolated from the fermented yoghurts collected from families in multiple cities of China and the intestinal contents from healthy piglets without pathogen infection and diarrhoea by our lab.

Sorry for the ambiguous and limited information, previously, more details had been added in Materials and methods section in the revised manuscript (see Line 482-493) (Manuscript with marked changes are related to “Related Manuscript File” in submission system). We gratefully appreciate your professional comments.

Line 482-493: “Lactic acid bacteria (LAB) and Enterococcus strains were isolated from 39 samples: 33 fermented yoghurts samples (collected from families in multiple cities of China, including Lanzhou, Urumqi, Guangzhou, Shenzhen, Shanghai, Hohhot, Nanjing, Yangling, Dali, Zhengzhou, Shangqiu, Harbin, Kunming, Puer), and 6 healthy piglet rectal content samples without pathogen infection and diarrhea in pig farm of Zhejiang province (Table 1). Ten isolates were randomly selected from each sample. De Man-Rogosa-Sharpe (MRS) with 2.0% CaCO_3_ (is a selective culture medium to favor the luxuriant cultivation of Lactobacilli) and Brain heart infusion (BHI) broths (Huankai Microbial, Guangzhou, China) were used for bacteria isolation and cultivation. Matrix-Assisted Laser Desorption Ionization-Time of Flight Mass Spectrometry (MALDI-TOF MS, Bruker Daltonik GmbH, Bremen, Germany) method was employed to identify of bacterial species with a confidence level ≥ 90% (He et al., 2022).”

Lines 129-133: A total of 290 bacterial strains were isolated and identified from 32 samples of the fermented yoghurt and piglet rectal contents collected across diverse regions within China using MRS and BHI medium, which consist s of 63 Streptococcus strains, 158 Lactobacillus/ Lacticaseibacillus Limosilactobacillus strains and 69 Enterococcus strains.

Sorry for the ambiguous information, we had carefully revised this section and more details had been added in this section (see Line 129-133). We gratefully appreciate your professional comments.

Line 129-133: “After identified by MALDI-TOF MS, a total of 290 bacterial isolates were isolated and identified from 33 fermented yoghurts samples and 6 healthy piglet rectal content samples. Those isolates consist of 63 Streptococcus isolates, 158 Lactobacillus/Lacticaseibacillus/Limosilactobacillus isolates, and 69 Enterococcus isolates (Figure 1A, Table 1).”

Main-point-2:As probiotics, Lacticaseibacillus rhamnosus has been widely studied. In fact, there are many commercially available products, and Lacticaseibacillus rhamnosus is the main bacteria in these products. There are also ATCC type strain such as 53103.I am sure the authors are also interested to know if P118 is better as a probiotics candidate than other commercially available strains. Also, would the mechanism described for P118 apply to other Lacticaseibacillus rhamnosus strains?It would be ideal if the authors could include one or two Lacticaseibacillus rhamnosus which are currently commercially used, or from the ATCC. Then, the authors can compare the efficacy and antibacterial mechanisms of their P118 with other strains. This would open the windows for future work.

We gratefully appreciate your professional comments and valuable suggestions. We deeply agree that it will be better and make more sense to include well-known/recognized/commercial probiotics as a positive control to comprehensively evaluate the isolated P118 strain as a probiotic candidate, particularly in comparison to other well-established probiotics, and also help assess whether the mechanisms described for P118 are applicable to other *L. rhamnosus* strains or lactic acid bacteria in general. Those issues will be fully taken into consideration and included in the further works. Nonetheless, the door open for future research had been left in Conclusion section (see Line 477-479) “Further investigations are needed to assess whether the mechanisms observed in P118 are strain-specific or broadly applicable to other L. rhamnosus strains, or LAB species in general.”.

**Recommendations for the authors:**

**Reviewer #2 (Recommendations for the authors):**
Minor comments:This reviewer appreciates the efforts from the authors to provide the details related to this work. In the meantime, the manuscript shall be written in a way which is easy for the readers to follow.

We had tried our best to revise and make improve the whole manuscript to make it easy for the readers to follow (e.g., see Line 27-30, Line 115-120, Line 129-133, Line 140-143, Line 325-328, Line 482-493, Line 501-502, Line 663-667, Line 709-710, Line 1003-1143). We gratefully appreciate your valuable suggestions.

For example, under the sections of Materials and Methods, there are 19 sub-titles. The authors could consider combining some sections, and/or cite other references for the standard procedures.

We gratefully appreciate your professional comments and valuable suggestions. Some sections had been combined according to the reviewer’s suggestions (see Line 501-710).

Another example: the figures have great resolution, but they are way too busy. The figures 1 and 2 have 14-18 panels. Figure 5 has 21 panels. Please consider separating into more figures, or condensing some panels.

We deeply agree with you that some submitted figures are way too busy, but it’s not easy for us to move some results into supplementary information sections, because all of them are essential for fully supporting our hypothesis and conclusions. Nonetheless, some panels had been combined or condensed according to the reviewer’s suggestions (see Line 1003-1024, Line 1056-1075). We gratefully appreciate your professional comments and valuable suggestions.

More minor comments:line 30: spell out "C." please.

Done as requested (see Line 29, Line 31). We gratefully appreciate your valuable suggestions.